# Pro-Inflammatory Cytokines Transactivate Glycosylated Cytokine Receptors on Cancer Cells to Induce Epithelial–Mesenchymal Transition to the Metastatic Phenotype

**DOI:** 10.3390/cancers17071234

**Published:** 2025-04-05

**Authors:** Leili Baghaie, David A. Bunsick, Emilyn B. Aucoin, Elizabeth Skapinker, Abdulrahman M. Yaish, Yunfan Li, William W. Harless, Myron R. Szewczuk

**Affiliations:** 1Department of Biomedical & Molecular Sciences, Queen’s University, Kingston, ON K7L 3N6, Canada; 16lbn1@queensu.ca (L.B.); davidbunsick68@gmail.com (D.A.B.); 2Faculty of Science, Biology (Biomedical Science), York University, Toronto, ON M3J 1P3, Canada; emilyn08@my.yorku.ca; 3Faculty of Arts and Science, Queen’s University, Kingston, ON K7L 3N9, Canada; 21ess18@queensu.ca (E.S.); 18yl210@queensu.ca (Y.L.); 4Faculty of Health Sciences, Queen’s University, Kingston, ON K7L 3N9, Canada; a.yaish@queensu.ca; 5ENCYT Technologies Inc., Membertou, NS B1S 0H1, Canada

**Keywords:** pro-inflammatory cytokines, glycosylated cytokine receptors, EMT, neuraminidase-1, oseltamivir phosphate

## Abstract

Many patients who undergo a potentially curative surgical removal of their primary cancerous tumor remain at high risk for disease recurrence. This can be a localized or metastatic recurrence, the latter of which is often fatal. Recent studies have highlighted the important role of pro-inflammatory cytokines released after surgical resection, especially TGFβ-1, IL-6, and HGF, in promoting epithelial–mesenchymal transition (EMT) and enhancing cancer cell metastasis potential. These cytokines increase mesenchymal marker expression, reduce epithelial markers, and activate glycosylated cytokine receptors on cancer cells, which are crucial for EMT and the development of metastatic phenotypes. This study demonstrates that the cytokine receptors activated by these distinct cytokines are regulated by the enzyme mammalian neuraminidase-1 (Neu-1), suggesting a highly conserved signaling mechanism that may represent a novel therapeutic target against human malignancies.

## 1. Introduction

Cytokines are cellular signaling molecules secreted by lymphocytes, macrophages, natural killer (NK) cells, mast cells, and stromal cells, forming a communication network within the immune system [1,2,3]. The interaction of a cytokine with its receptor induces transcriptional changes that alter the expression of adhesion molecules and chemokine receptors, signal immune cell recruitment, aid in cellular proliferation and differentiation, and can trigger apoptosis [3]. The term cytokine covers an array of signaling molecules, most commonly interleukins (IL), interferons (IFNs), tumor necrosis factors (TNFs), colony-stimulating factors (CSFs), chemokines, and certain growth factors such as the transforming growth factor family (TGF) [1]. Cytokines are of particular concern in metastasis, as they can activate dormant cancer stem cells and foster metastasis [4]. The tumor microenvironment (TME) is highly inflamed with a landscape of cytokines, typically a result of oncogene activation or proteins released from necrosis, that can result in cytokine secretion and tumor invasiveness [5].

The epithelial-to-mesenchymal transition (EMT) is an important process that can enhance the ability of a cancer cell to migrate and colonize tissue. These processes are essential to successful metastatic spread. EMT is characterized by a change in cell morphology from a cobblestone epithelial-type shape to an elongated spindle-shaped fibroblast-like appearance [6,7]. The multifunctional cytokine TGF-β is known to be a crucial driver of EMT in various cancer cells [7]. TGF-β can transduce signals via a single-pass transmembrane Ser/Thr kinase receptors and co-receptors, which have glycosylated extracellular domains [8]. Extracellular and intracellular signaling through TGF-β is intricately regulated, involving the glycosylation of cell surface TGF-β-binding proteins. These changes in glycosylation are of critical importance for the cellular responses induced by TGF-β, including the EMT. Most of the cytokine receptors are cell surface glycosaminoglycans that belong to the G-protein-coupled receptors (GPCRs) family [9].

We have conceptualized that the wound-healing response triggered by cancer surgery may facilitate the growth and metastatic potential of any surviving cancer cell population [10]. Recently, Baghaie et al. [11] investigated plasma samples from surgical breast, prostate, and colorectal cancer patients obtained from sixty-eight cancer patients who had not received treatment before surgery or adjuvant therapy until at least four weeks post-surgery. The levels of chemokines, plasma cytokines, and growth factors were simultaneously quantified and profiled using multiplexed immunoassays for eight time points sampled per patient. After surgery, there was a drastic short-term shift in the expression levels of pro-inflammatory and angiogenic molecules and cytokines. A rapid and significant increase in circulating levels of plasma of hepatocyte growth factor (HGF), interleukin-6 (IL-6), placental growth factor (PLGF), and matrix metalloproteinase-9 (MMP-9) was noted after surgery. Interestingly, with the rise in these molecules, there was a significant concomitant drop in transforming growth factor-1 (TGF-1), platelet-derived growth factor (PDGF-AB/BB), insulin-like growth factor-1 (IGF-1), and monocyte chemoattractant protein-2 (MCP-2). Notably, each analyte of plasma within 1–2 weeks after surgery was normalized to baseline levels, suggesting that surgical intervention alone was responsible for these changes. The rapid effects of surgical tumor removal on the expression of a variety of cytokines important in inflammation and angiogenesis raise important questions about the approach of current clinical practice of delaying adjuvant chemotherapy after surgery to allow time for wound healing. The wound-healing process triggered by cancer surgery may likely enhance the ability of a residual cancer cell population to proliferate and metastasize [10,11].

Based on previous work, a critical role of neuraminidase-1 (Neu-1) in the regulation of a number of tyrosine kinase receptors has been implicated in cancer cell proliferation and metastasis [12,13]. We further explored whether this same mechanism-signaling process is present in the cognate cytokine receptors for IL-6, HGF, and TGF-beta. Cytokine receptor binding is proposed here involving neuromedin-B (NMBR) GPCR-MMP-9-Neu-1 signaling crosstalk based on our previous reports of receptor tyrosine kinase (RTK) activation. The binding of a cytokine to its receptor, which is a receptor tyrosine kinase (RTK), is proposed to induce a conformational change associated with NMBR GPCR, which in turn activates MMP-9 through the Gα subunit. This activation induces Neu-1 by releasing the elastin-binding protein (EBP) from its complex with Neu-1 and protective protein cathepsin A (PPCA). Neu-1 is now able to cleave α-2,3 sialic acid from the terminal residue of the RTK, relieving steric hindrance, which in turn internalizes the receptor’s cytokine-dependent signal. The inhibition of Neu-1 by oseltamivir phosphate (OP) was demonstrated to decrease the downstream signaling of the receptor–ligand binding implicated in cancer progression [14]. OP’s inhibition of Neu-1 prevents the cleavage of α-2,3 sialic acid and prevents receptor activation [13]. In mouse models of human pancreatic cancer, OP decreased tumor growth with no side effects [14]. The inhibition of NMBR using BIM-23127 and the inhibition of MMP9 with MMP9-i have demonstrated similar functions in cancer [14] and oseltamivir and peramivir on SARS-CoV-2 infection progression [15].

Cytokines have been reported to regulate epithelial–mesenchymal transition (EMT), a key role played in metastasis [16,17,18,19,20,21]. O’Shea et al. [22] reported on the therapeutic potential of oseltamivir phosphate (OP) that can reverse EMT. This reversal, known as mesenchymal-to-epithelial transition (MET), reduces chemotherapy resistance and enhances the efficacy of existing treatments [22]. OP has also been reported to significantly decrease the activity of chemically resistant PANC1 cells [22]. Furthermore, OP was reported to increase the epithelial marker E-cadherin and decrease the mesenchymal marker expression of N-cadherin and VE-cadherin [22,23]. Skapinker et al. [24] recently reported that OP treatment on day 35 of tumor-bearing mice maintained enhanced levels of angiostatic and fibrogenic cytokines. These data suggest that OP may have a significant role to play in regulating cytokine signaling in tumor development.

The NMBR-MMP9-Neu-1 crosstalk leads to a conformational change(s) in the receptor, leading to ERK/MAPK activation mediated through signaling from a zinc-finger factor snail [25]. The snail transcriptional factor has been previously shown to induce the transcription and expression of MMP-9, which is linked to epithelial–mesenchymal transition (EMT) and neovascularization in the tumor microenvironment [25].

Cytokine-dependent receptor signaling also contributes to downstream signaling that supports the synthesis of the inflammatory and profibrotic cytokines TGF-β and IL-6 [26,27], which are also expressed in most inflammatory cells, where they induce IL-6 secretion from macrophages; IL-1β, IL-8, IL-4, and VEGF from mast cells; and TNF-α from neutrophils [25,28]. In contrast, the adenosine A2BAR triggers an anti-inflammatory response by inhibiting TNF-α and IL-1β production in neutrophils [28].

To this end, we hypothesized that the binding of cytokines TGF-β1, IL-6, and HGF to their respective receptors induces cancer cells to adopt a mesenchymal metastatic phenotype through the Neu-1-MMP9-GPCR crosstalk signaling platform. Targeting Neu-1 with OP is proposed to inhibit this cytokine-induced metastatic transformation.

## 2. Materials and Methods

### 2.1. Cell Lines

PANC-1 (human, epithelial morphology, ATCC CRL-1469TM, Manassas, VA, USA) isolated from pancreatic duct epithelioid carcinoma, SW620 (human, epithelial morphology, ATCC CCL-227TM) isolated from the large intestine and colon of colorectal adenocarcinoma, and MCF-7 (human, epithelial morphology, ATCC HTB-22TM) isolated from mammary adenocarcinoma were obtained from the American Type Culture Collection (ATCC; Manassas, VA, USA). RAW-Blue macrophages were also acquired (InvivoGen, San Diego, CA, USA). Cells were grown in culture media containing Dulbecco’s Modified Eagle’s Medium (DMEM) (Gibco, Rockville, MD, USA) supplemented with 10% fetal bovine serum (FBS) (HyClone, Logan, UT, USA) and 5 µg/mL plasmocin (InvivoGen, San Diego, CA, USA). Cells were maintained in an incubator with 5% CO_2_ at 37 °C until they reached 75% confluence. The cells were passaged 2–3 times before use in the experiment. Cancer cell line selection was based on the dramatic changes in levels of TGFβ-1, IL-6, and HGF noted in colorectal and breast cancer patients following surgical resection [11]. Pancreatic cancer was additionally selected as it is an aggressive and highly metastatic cancer.

### 2.2. Cytokines

The three cytokines selected for this study were chosen based on their significant alterations observed in the plasma profiles of breast, colorectal, and prostate cancer patients after surgical resection, as reported by Baghaie et al. [11]. The concentrations of these cytokines resembled the peak levels observed in the same study: transforming growth factor β-1 (TGFβ-)(Abcam Inc., Cambridge, MA, USA) at 4.0 × 10^−3^ µg/mL, interleukin-6 (IL-6) (Sigma-Aldrich, St. Louis, MI, USA) at 4.1 × 10^−5^ µg/mL, and hepatocyte growth factor (HGF) (Sigma-Aldrich, St Louis, MI, USA) at 5.97 × 10^−4^ µg/mL. The cytokines were diluted in media without FBS to prevent any confounding interactions of cells with the rich source of proteins and growth factors in FBS.

### 2.3. Antibodies

#### 2.3.1. Immunofluorescence 

Primary mouse monoclonal IgG antibodies for ALDH-1 (sc-374076), E-cadherin (sc- 8426), N-cadherin (sc-8424), and vimentin (sc-6260) were obtained from Santa Cruz Biotechnology (Dallas, TX, USA). Goat anti-mouse IgG secondary antibody Alexa Fluor^TM^ 488 (A10667; Invitrogen, Thermo Fisher Scientific Inc., Waltham, MA, USA) was used as a fluorophore secondary antibody and negative control. Alternatively, Alexa Fluor^TM^ 594-conjugated antibodies for E-cadherin (sc-8426 AF594), N-cadherin (sc-8424 AF594), and vimentin (sc-6260 AF594) and stem cell markers (CD24 and CD44) were acquired from Santa Cruz Biotechnology (Dallas, TX, USA). Antibodies were made up of 5% BSA/PBST and incubated at 4 °C overnight. The slides of stained cells were washed with PBS 10× for 5 min each, incubated with the appropriate secondary antibodies, and incubated with DAB (3,30-diaminobenzidine, Vector Laboratories, Inc., Burlingame, CA, USA). All antibodies were diluted (1:50) based on the amounts specified by the manufacturers’ protocols.

#### 2.3.2. Colocalization

A mouse monoclonal IgG TGFβ-R1 Alexa Fluor^TM^ 594-conjugated antibody (sc-518018) and a mouse monoclonal IgG Neu-1 Alexa Fluor^TM^ 488-conjugated antibody (sc-166824) were acquired from Santa Cruz Technologies (Dallas, TX, USA). A mouse IL-6R alpha Alexa Fluor^TM^ 594-conjugated antibody (FAB1830T) and a human HGFR/c-MET Alexa Fluor^TM^ 594 conjugated antibody (FAB3582T) were acquired from R&D Systems (Minneapolis, MN, USA). All antibodies were diluted based on the amounts specified by the manufacturers’ protocols.

### 2.4. Inhibitors

Oseltamivir phosphate (OP) (batch no. MBAS20014A, Solara Active Pharma Sciences Ltd., New Mangalore-575011, Karnataka, India), a broad-range sialidase inhibitor that inhibits Neu-1, was used at 300 µg/mL. An MMP9 inhibitor (MMP9i) (GC44237; GLPBIO, Montclair, CA, USA) was used at 26 ng/mL. BIM 23127 (GC11620; GLPBIO, Montclair, CA, USA), an antagonist of the neuromedin B receptor (NMBR) of the G-protein-coupled receptors (GPCRs) family, was used at a concentration of 25 ng/mL.

### 2.5. Immunofluorescence Protocol

PANC-1, SW620, MCF-7, and RAW-Blue macrophage cells were grown in a culture medium containing 10% FBS at a density of 200,000 cells/well on 12 mm glass coverslips in a 24-well plate and incubated at 37 °C until 75% confluence was reached. The adhered cells were then treated with cytokines diluted in DMEM without FBS (see Section 2.2 Cytokines), and control cells were incubated with FBS-free media for 24 h. Cells were washed and then fixed with 4% paraformaldehyde (PFA) overnight in a 4 °C fridge. The following day, cells were permeabilized with 0.2% TritonX 100 for 5 min at 37 °C and blocked with 4% Bovine Serum Albumin (BSA) in 0.1% Tween20-TBS solution for 20 min at room temperature. Following blocking, the cells were treated with a 1:50 ratio of primary mouse monoclonal IgG antibody of interest and left in a 4°C fridge for 24 h. Following incubation, cells were washed with PBS-T and then treated with a 1:1000 ratio of an Alexa Fluor^TM^ 488-conjugated secondary antibody. For negative control, cells lacking a primary antibody were incubated with only the tagged secondary antibody of the same concentration. Coverslips with attached cells were inverted onto a slide with VECTASHIELD DAPI fluorescent mounting medium (VECTH1500, MJS BioLynx Inc., Brockville, ON, Canada). Slides were immediately visualized using Zeiss M2 epi-fluorescent microscopy (20× objective magnification, 488 nm, Carl Zeiss Canada Ltd., Toronto, ON, Canada). Fluorescent density was measured based on background means, image means, and pixel values using Corel Photo-Paint X8 using the following equation:Density = (background mean − image mean) × pixels.

### 2.6. Colocalization Protocol

PANC-1, SW620, and MCF-7 cells were grown in a culture medium containing 10% FBS at a density of 200,000 cells/well on 12 mm glass coverslips in a 24-well plate and incubated at 37 °C until 75% confluence was reached. Cells were then serum-starved for 24 h and subsequently fixed with 4% PFA overnight in a 4 °C fridge. Following fixation, cells were washed and permeabilized with TritonX 100 for 5 min at 37 °C, then blocked with 4% BSA in 0.1% Tween20-TBS solution for 24 h in a 4°C fridge. Cells were then treated with 150 µL of the Alexa Fluor594-conjugated antibodies for the cytokine receptors of interest and 150 µL of the Alexa Fluor488-conjugated Neu-1 antibody and left in a cold room on a rocker for 48 h. To account for non-specific background fluorescence (negative control), cells were treated only with an IgG secondary antibody conjugated with Alexa Fluor488 or Alexa Fluor594. Coverslips with attached cells were inverted onto a slide with VECTASHIELD DAPI fluorescent mounting medium (VECTH1500, MJS BioLynx Inc. Brockville, ON, Canada). Slides were immediately visualized using Zeiss M2 epi-fluorescent microscopy (40× objective magnification, Carl Zeiss Canada Ltd., Toronto, ON, Canada). The Pearson correlation coefficient demonstrating the colocalization of the two fluorescent-antibody channels was determined using AxioVision software, Rel. version 4.6. A correlation value of 0.5 to 1.0 between the two variables indicates a significant positive relationship.

### 2.7. Sialidase Activity Assay

PANC-1, SW620, and MCF-7 cells were grown in a culture medium containing 10% FBS at a density of 200,000 cells/well on 12 mm glass coverslips in a 24-well plate and incubated at 37 °C until 75% confluence was reached. The adhered cells were then serum-starved for 24 h at 37 °C. To measure neuraminidase activity, 2-(4-methylumbelliferyl)-α-D-N-acetylneuraminic acid (98% pure 4-MUNANA; Biosynth International Inc., Itasca, IL, USA), a sialidase substrate with excitation at 365–380 nm and emission at 445–454 nm, was used. In each experimental condition, 3 µL of 4-MUNANA was added. Control cells received only 4-MUNANA, while cytokine-treated cells additionally received 3 µL of each cytokine in media without FBS. Inhibitor-treated cells had 3 µL of each inhibitor added along with cytokines and 4-MUNANA at the same time. Fluorescent and phase contrast images were captured using Zeiss M2 epi-fluorescent microscopy (20× objective magnification, Carl Zeiss Canada Ltd., Toronto, ON, Canada) with the compounds and 4-MUNANA. The mean fluorescence intensity of 50 different points surrounding the cell was quantified using Image J software.

### 2.8. AlamarBlue Viability Assay and Metabolic Activity

Cytokine-induced viability or cytotoxicity was determined using the AlamarBlue assay, as previously described [14]. The PANC-1 and SW620 cells were seeded at 20,000 cells per well in flat-bottom 96-well plates. Cells were incubated at 37 °C at 5% CO_2_ for 24h before being treated with cytokines or an untreated vehicle control. Cytotoxicity was determined using the AlamarBlue reagent. A total of 10 µL of AlamarBlue reagent was added to every 100 µL of supernatant and incubated for 4 h at 37 °C and 5% CO_2_. The absorbance was recorded using a fluorophore excitation (ex 560 nm; em 590 nm). The results were calculated by subtracting fluorescence from the blank media control and compared to the untreated control.

### 2.9. Statistical Analysis

All statistical analyses were conducted using GraphPad Prism 10 software and are presented as mean ± standard error of the mean (SEM). Comparisons between groups from two to three independent experiments were made by one-way analysis of variance (ANOVA) at 95% confidence using the uncorrected Fisher’s LSD multiple comparisons test with 95% confidence. Asterisks denote statistical significance.

## 3. Results

### 3.1. TGFβ-1, IL-6, and HGF Binding Respective Cytokine Receptors Induce Sialidase Activity

We reported a novel neuraminidase-1 (Neu-1) and matrix metalloproteinase-9 (MMP-9) crosstalk in alliance with the GPCR neuromedin B, which is essential for ligand-induced RTK activation and cellular signaling [13]. It was important to ascertain whether the binding of the cytokines to their ligands on the cell surface could induce downstream signaling through conformational changes in the signaling paradigm depicted in Figure 1. Here, live PANC-1 (Figure 2), SW620 (Figure 3), and MCF7 (Figure 4) cancer cells were added to TGFβ-1 (4.0 × 10^−3^ µg/mL), IL-6 (4.1 × 10^−5^ µg/mL), HGF (5.97 × 10^−4^ µg/mL), or untreated together with sialidase substrate 4-MUNANA (emission 450 nm, excitation 365 nm) for 1 min. Phase contrast images showed the presence of cells, while the fluorescent images measured the sialidase activity (blue color). In all three cell lines tested, the untreated PANC-1 cells had very low levels of sialidase activity compared to TGFβ-1, IL-6, and HGF-treated cells (*p* < 0.0001) (Figure 2). Together, these results demonstrate that sialidase activity is only induced by cytokine binding to its receptor. Similar results were found for colorectal SW620 (Figure 3) and breast MCF7 (Figure 4) cancer cell lines.

To test the hypothesis that cytokine receptor binding is implicated in the Neu-1-MMP-9-GPCR crosstalk, as depicted in Figure 1, we investigated the inhibition of cytokine-induced Neu-1 sialidase activity with oseltamivir phosphate (OP), as well as with the other components of the signaling paradigm, such as MMP9 (MMP9i) and NMBR (BIM23). In all cell lines tested, the addition of OP, MMP9i, and BIM23 to cytokine-treated cells simultaneously blocked the sialidase activity induced by the cytokines (Figure 2D,H,I). While the inhibition was highly significant with all three inhibitors (*p* < 0.0001), OP decreased the sialidase activity the greatest and lower than the untreated control (Figure 2D,H,I). It is noteworthy that the inhibitors alone did not have cytotoxic effects on the cells (Figure 2M). In addition, all of the phase contrast images showed confluent live cell images. These findings support the hypothesis that the NMBR-MMP-9-Neu1 signaling platform regulates glycosylated cytokine receptors, as depicted in Figure 1.

### 3.2. Cytokine Receptors Colocalize with Neu1 on the Cell Surface of Naïve Unstimulated PANC-1, SW-620, and MCF7 Cells

The data depicted in Figure 2, Figure 3 and Figure 4 demonstrate that the cytokine binding to their respective receptors induced Neu-1 sialidase activity. Here, we investigated whether the cytokine receptors form a complex with the signaling platform of Neu1 and MMP-9 crosstalk in alliance with the NMBR GPCR on the cell surface. As depicted in Figure 5, Figure 6 and Figure 7, cells were treated with equal amounts of Alexa Fluor™ 594-conjugated monoclonal antibodies for the cytokine receptors TGFβR, IL-6R, and HGFR/c-MET and the Alexa Fluor™ 488-conjugated monoclonal anti-Neu-1 antibody. The colocalization was quantified using a Pearson correlation coefficient (R-value; 0.5 < r < 0.59 = moderate positive correlation; 0.6 < r < 0.89 = strong positive correlation; r > 0.9 = near perfect correlation).

For PANC-1 cells, the cytokine receptors and Neu-1 had a strong positive correlation (TGFβ-1R: r = 0.72 ± 0.06; IL-6R: r = 0.69 ± 0.10; HGFR/c-MET: r = 0.74 ± 0.01) (Figure 5). In contrast, the SW620 cells demonstrated a moderate positive correlation (TGFβ-1R: r = 0.55 ± 0.00; IL-6R: r = 0.57 ± 0.03; HGFR/c-MET: r = 0.60 ± 0.00) (Figure 6). Interestingly, Matsuo et al. [30] demonstrated that SW620 cells have a higher level of IL-6 receptor secretion than the other colorectal cancer cell lines. Noteworthy, the concentrations of IL-6R in each cell supernatant were measured by an enzyme-linked immunosorbent assay on the third day of culture.

For MCF-7 cells, the cytokine receptors markedly colocalized with Neu-1 as represented by their high Pearson correlation coefficients (TGFβ-1R: r = 0.81 ± 0.05; IL-6R: r = 0.92 ± 0.01; HGFR/c-MET: r = 0.89 ± 0.06) (Figure 7).

These findings demonstrate that for PANC-1 and MCF-7 cells, the cytokine receptors and Neu-1 have a high colocalization. The SW620 cells showed a moderate but not as strong receptor-Neu-1 colocalization as with the other two cell lines, but they still exhibited a functional interaction. These results support the sialidase activities with the specific inhibitors of the signaling platform, as depicted in Figure 1.

### 3.3. Effects of TGFβ-1, IL-6, and HGF on the Viability and Metabolic Activity of PANC-1, SW620, and MCF-7 Cancer Cells

Metabolic rewiring and metastatic cascades are intertwined cooperatively to promote cancer metastasis [31,32]. Different metabolites such as amino acids at secondary organ sites [33], lipids [34], and cytokines in the tumor microenvironment [35] generated by cancer cells [24] can influence the metastatic cascade, encompassing epithelial–mesenchymal transition (EMT), the survival of cancer cells in circulation, and metastatic colonization at distant sites. The molecular mechanisms that underlie the prometastatic niche effect of tumor-derived metabolites can be epigenetic deregulation, the induction of matrix metalloproteinases (MMPs), and the promotion of cancer stemness [31,32]. The metastatic signaling regulates the rate-limiting metabolic enzymes that generate the prometastatic metabolites, thereby promoting the metastasis cascade. The specific metabolic alterations in cancer development may be a metabolic functional driver of tumor growth and progression, and as a result, dysregulated metabolic pathways have become attractive targets for cancer therapeutics [36]. Here, we investigated the metabolic activity of PANC-1, SW620, and MCF-7 cancer cells following treatment with TGFβ-1, IL-6, and HGF at predetermined concentrations.

An AlamarBlue cell viability assay was used to ascertain the impact of cytokines on cell viability and metabolic activity. This assay is used widely to investigate cell proliferation and cellular metabolic activity using the oxidation–reduction indicator resazurin (absorbance 570 nm). The AlamarBlue reagent is an indigo-colored, non-toxic resazurin-based solution that acts as an indicator of cell health by using the reducing power of living cells to measure viability quantitatively. Upon entering living cells, the resazurin is reduced to resorufin. This red compound is highly fluorescent and provides accurate time-course measurements of the metabolic activity of healthy cells with high sensitivity and linearity and no cell lysis [37].

The data depicted in Figure 8A–C reveal that TGFβ-1, IL-6, and HGF significantly (*p* < 0.0404) increased the cell viability of PANC1 and SW620 cells, whereas TGFβ-1 significantly (*p* < 0.0065) increased MCF-7 cell viability. In contrast, IL-6 significantly (*p* < 0.0006) decreased it, and HGF had no significant impact on MCF-7 cell viability when compared to the untreated control. These results indicated that these cytokines may have a pro-survival role for specific cancer cells, such as pancreatic and colorectal cancer cells, which aid in their invasion and migration properties.

Of interest, we also examined the impact of growth factors in culture media, such as fetal bovine serum (FBS). In all three cell lines, the addition of FBS significantly increased cell viability, with SW620 cells having the most significant and dramatic rise (SW620, *p* < 0.0001; PANC-1 and MCF-7, *p* < 0.0404) (Figure 8A–C). Notably, diluting cytokines in media without FBS further establishes their sheer impact on cell viability. To this end, we determine the impact of OP on cytokine-induced cell viability. We first determined the best treatment timing with OP using an AlamarBlue assay measuring overall cell viability at 5 min, 10 min, and 15 min. The findings of these preliminary experiments determined that a treatment time of 15 min with OP prior to the addition of cytokines had the best starting cell viability for these cells (Figure 8D–F). In all cell lines, the co-treatment of TGFβ-1 with OP significantly decreased (*p* < 0.0065) cell viability to slightly below untreated levels (Figure 8D). Similarly, with the co-treatment of IL-6 with OP in PANC-1 and SW620 cells, a significant decrease was seen compared to the cell viability of IL-6 alone (*p* < 0.0404); however, there was an insignificant difference in cell viability with MCF-7 (Figure 8E). Concerning HGF and OP, all three cell lines demonstrated that a combination of OP with HGF significantly decreases (*p* < 0.0065) the cell viability compared to HGF treatment alone (Figure 8F). These findings demonstrate that while OP decreases cell viability, it does not induce cell death, providing us with an opportunity to evaluate its applicability in cancer treatment.

### 3.4. TGFβ-1, IL-6, and HGF Induce the EMT Phenotype in PANC-1, SW620, and MCF-7 Cancer Cells

To determine the impact of cytokines on epithelial–mesenchymal transition (EMT), PANC-1, SW620, and MCF-7 cancer cells were treated with TGFβ-1 (4.0 × 10^−3^ µg/mL), IL-6 (4.1 × 10^−5^ µg/mL), or HGF (5.97 × 10^−4^ µg/mL) for 24 h. Control cells were left untreated but serum-starved (media without FBS) for 24 h. Using mouse monoclonal primary antibodies directed to the cell surface markers of EMT and anti-mouse fluorescently tagged (Alexa Fluor™ 488) secondary antibodies, immunofluorescent images were collected and analyzed. For PANC-1 cells, the expression of mesenchymal markers ALDH-1, vimentin, and N-cadherin significantly increased (*p* < 0.0069) following the cytokine treatments (Figure 9). Notably, vimentin is an intermediate filament that is upregulated during epithelial–mesenchymal transition (EMT) and serves as a marker for cancer progression, metastasis, and stem-like cell properties [38]. Alcohol dehydrogenase-1 family member A1 (ALDH1A1) is a marker of chemoresistance and cancer stem-like properties. ALDH1A1, a key aldehyde dehydrogenase isoform, is linked to cancer stem cells (CSCs) and epithelial–mesenchymal transition (EMT) and used as an EMT marker for metastasis-initiating cells (MICs) [39]. The ALDH1 enzyme is also involved in the oxidation of retinol to retinoic acid and is an essential process for stem cells in their early differentiation [40]. In contrast to the mesenchymal markers, the epithelial marker E-cadherin was significantly downregulated (*p* < 0.0019) with the cytokine treatments (Figure 9).

SW620 cells demonstrated a similar pattern, in which ALDH-1 and vimentin expression were significantly upregulated with TGFβ-1, IL-6, and HGF treatment compared to the untreated control (*p* < 0.028) (Figure 10B,C). Conversely, cell surface levels of E-cadherin decreased (*p* < 0.0031) in cytokine-treated cells (Figure 10A), as seen in pancreatic cancer cells. Notably, N-cadherin was not measured because previous reports have established that SW620 cells do not express this cell surface marker [41].

The MCF-7 cells had a similar upregulation of the mesenchymal marker N-cadherin (*p* < 0.0059) following cytokine treatments (Figure 11). A significant decrease in E-cadherin was also seen in TGFβ-1, IL-6, and HGF-treated cells (Figure 11). As with SW620 cells not expressing N-cadherin, vimentin was not quantified for MCF-7 cells as they are reported as not being expressed by these cells [42].

Together, these findings suggest that TGFb-1, IL-6, and HGF cytokines induce an EMT transition to the mesenchymal phenotype.

### 3.5. The Effect of TGFβ-1, IL-6, and HGF on CD44 and CD24 as Markers for Identifying and Characterizing Cancer Stem Cells (CSCs) to Metastasize

CD44 and CD24 are cell surface proteins, and their expression patterns are used to identify and characterize cancer stem cells (CSCs) [43]. A high CD44/CD24 ratio, particularly the CD44 high/CD24 low phenotype, is characteristically associated with a higher proportion of CSCs within a tumor [44]. A high CD44/CD24 ratio is also linked to the ability of cancer cells to metastasize [44]. To this end, we investigated the CD44/CD24 ratio of PANC-1 and SW620 cells following treatment with TGFβ-1, IL-6, and HGF cytokines. Here, we followed the same cytokine protocol treatments as depicted in Figure 7, Figure 8 and Figure 9. Interestingly, TGFβ-1, IL-6, and HGF induced a high CD44/CD24 ratio in PANC-1 cells (Figure 12). For SW620 cells, only TGFβ-1 and IL-6 induced this high ratio, which is indicative of transforming these cells toward metastatic potential. In contrast, HGF did not affect SW620 cells compared to the untreated control cells.

### 3.6. Cells Do Not Express Markers of EMT Treated with Co-Combination of OP and Cytokines

PANC-1, SW620, and MCF-7 cells were treated with coculture of 300 µg/mL OP for 15 min before the addition of either TGFβ-1 (4.0 × 10^−3^ µg/mL), IL-6 (4.1 × 10^−5^ µg/mL), or HGF (5.97 × 10^−4^ µg/mL). Cells were stained for immunofluorescence following the same protocol as Section 2.5, with a slight adjustment of using the Alexa Fluor 488™ conjugated antibody for the cell surface markers instead of the primary and secondary antibodies previously used. While DAPI nuclear staining showed intact nuclei of cells, there was no measurable fluorescence with E-cadherin, N-cadherin, or vimentin, suggesting a potential degradation of EMT cell surface markers (Figure 13).

### 3.7. Macrophage Cells Express a Hybrid of EMT Markers When Treated with Cytokines

Hybrid EMT refers to a state where cancer cells retain some epithelial characteristics while simultaneously acquiring mesenchymal features rather than undergoing a complete transition to a fully mesenchymal state [45]. RAW-Blue macrophages were treated with cytokines following the same protocol as in Figure 8, Figure 9 and Figure 10, using primary and secondary (Alexa Fluor™ 488) antibodies. An analysis of the immunofluorescent results found that a mixture of both epithelial and mesenchymal markers is expressed, suggesting a potential hybrid state of EMT (Figure 14A–C). HGF-treated cells had a significantly increased expression (*p* = 0.0038) of E-cadherin with a simultaneous significant upregulation (*p* = 0.0024) of vimentin. Similarly, the IL-6 treatment demonstrated a significant decrease in vimentin (*p* = 0.0122) paired with a significant increase in N-cadherin (*p* = 0.0149). These results contradict the EMT immunofluorescence data in the cancer cells previously discussed; however, they suggest further investigation may uncover novel information about macrophages and other immune cells’ role in cellular motility.

## 4. Discussion

In the present study, we first demonstrate the role that the cytokines TGFβ-1, IL-6, and HGF play in the epithelial–mesenchymal transition (EMT). This process is strongly linked to the increased metastatic potential of cancer cells. We noted a statistically significant upregulation of mesenchymal markers N-cadherin, vimentin, and ALDH-1 paired with a decreased expression of epithelial E-cadherin following treatment with the cytokines, indicative of a transition to the mesenchymal phenotype. These findings are in line with previous work in which these cytokines contribute to the induction of EMT in various cancer cells [17,46,47,48,49,50].

Our investigation noted a strong link between these distinct cytokines’ receptor signaling and the Neu-1-MMP-9-GPCR crosstalk. Results from the colocalization assay of neuraminidase-1 (Neu-1) with TGFβR, IL-6R, and HGFR/c-MET demonstrated a strong colocalization between them in PANC-1 and MCF-7 cells and a moderate positive correlation in SW620 cells. These findings suggest that the downstream signaling of these cytokines with their receptors may be dependent on the signaling paradigm proposed (see Figure 1), particularly Neu-1. Our lab previously noted that the binding of epidermal growth factor (EGF) to its receptor, part of the receptor tyrosine kinase (RTK) family, induces conformational changes in the GPCR [13,25]. This change, in turn, activates MMP-9 through the Gα subunit, which releases the elastin-binding protein (EBP) from its complex with Neu-1 and protective protein cathepsin A (PPCA). This allows Neu-1 to cleave α-2,3 sialic acid from the terminal residue of the RTK, relieving steric hindrance, which activates downstream signaling from the cytokine receptor binding. As such, this study found significant increases in neuraminidase activity following the binding of the TGFβ-1, IL-6, and HGF to their receptors (Figure 2). This increased sialidase activity only occurred when cells were stimulated with cytokines and were found to be decreased back to untreated levels when inhibited with oseltamivir phosphate (OP), MMP9i, and BIM23. Together with the results from the colocalization assay, these findings suggest that the cytokines studied here activate their downstream effects via the Neu-1-MMP9-GPCR receptor signaling platform. The extent of inhibition was strongest with OP, an inhibitor of the cytokine receptor colocalized with Neu-1.

We previously published a study showing that aspirin can inhibit EGFR signaling by its ability to target mammalian Neu-1 [51]. We propose that the ability of aspirin to inhibit this signaling process is critical to its anti-cancer effects. Intriguingly, a recent study showed that treatment with aspirin after the surgical resection of colorectal cancer reduced the risk of cancer recurrence by 50% in patients harboring mutations in PI3K, which is downstream of the EGFR receptor [52]. To put this result in perspective, chemotherapy given after surgery rarely has this potentially curative effect on reducing the risk of the recurrence of cancer after surgery. Furthermore, we have shown that treatments targeting Neu-1 were very effective at disrupting cancer cell metastasis in preclinical animals [14,53]. These studies also demonstrated that treatment with OP reduced the expression of both markers of EMT and stem cell enrichment, confirming at the cellular level the likely mechanism behind the anti-metastatic effects of OP.

While this present study primarily concentrated on the interaction of cancer cells with cytokines, the interaction of these cytokines with immune cells was also of interest, particularly considering their role in the wound-healing response post-surgical resection. Following the same protocol as the immunofluorescence of cancer cells, RAW-Blue macrophages were treated with either TGFβ-1 (4.0 × 10^−3^ µg/mL), IL-6 (4.1 × 10^−5^ µg/mL), or HGF (5.97 × 10^−4^ µg/mL) and stained with E-cadherin, N-cadherin, and vimentin (Figure 14). The findings of this study were very different than what was noted in the cancer cells. In HGF-treated cells, a significant increase in both the epithelial marker E-cadherin and mesenchymal marker vimentin was noted. Furthermore, the other mesenchymal marker, N-cadherin, significantly decreased with the same cytokine treatment. IL-6-treated cells also demonstrated this contradiction, where a significantly lower expression of vimentin and significantly increased expression of N-cadherin were observed.

Since macrophages are not epithelial, the concept of EMT does not directly apply. Instead, macrophages exhibit high plasticity that results in changes in their adhesion, motility, and invasiveness in response to stimuli such as cytokines [54]. The pro-inflammatory M1 macrophage phenotype is often activated by IFN-γ and lipopolysaccharide (LPS), while its counterpart, the anti-inflammatory M2 macrophages, can be activated by IL-4 and IL-13 [54,55,56]. Recently, the polarization of macrophages into the M2 phenotype has been further differentiated into subgroups, in which macrophages that TGFβ stimulates are called the M2c macrophages [55,56]. This subtype is linked to tissue repair, angiogenesis, and tumor promotion through increased immunosuppressive function [54,56]. These M2c macrophages also downregulate pro-inflammatory cytokines such as TNFα, IL-6, and IL-12 [55]. One of the proposed mechanisms is through snail, a downstream TGFβ-dependent transcription factor [57].

As such, the non-significant change in E-cadherin and N-cadherin and the significantly lower levels of vimentin in TGFβ-1-treated RAW-Blue macrophages are indicative of the anti-inflammatory M2c environment. As vimentin is a marker associated with tissue remodeling and wound healing, the decline may also explain the decreased wound-healing response with cells exposed to the cytokines.

Moreover, the overexpression of E-cadherin in RAW-Blue macrophages has been shown to reduce the secretion of inflammatory cytokines like IL-6 and TNFs [58]. The significant overexpression of E-cadherin with HGF treatment may induce an anti-inflammatory environment, which would account for the increased vimentin and reduced wound-healing response observed. The exact role of N-cadherin in macrophages remains to be elucidated. However, its upregulation in IL-6-treated macrophages may suggest increased migratory power. Together, the findings of the hybrid expression of EMT markers in RAW-Blue macrophages imply that the signaling from cytokines after surgical resection could impede immune recruitment and transform macrophages into an anti-inflammatory state. This phenomenon may contribute to immune suppression that allows cancer cells to become highly invasive and metastatic.

## 5. Conclusions

The findings of this study demonstrate that distinct cytokines released after the surgical resection of a primary tumor have a proliferative and invasive impact on pancreatic, colorectal, and breast cancer cells. They play a fundamental role in increasing the expression of mesenchymal markers while reducing epithelial markers, allowing cancer cells to grow and become mobile. These properties are predicted to foster the metastatic spread of a surviving cancer cell population after surgical removal of the primary tumor.

The mechanism we propose is through crosstalk between Neu-1-MMP-9-GPCR. Here, we show that the receptors for TGFβ-1, IL-6, and HGF colocalize with neuraminidase-1 and that the inhibition of Neu-1 using oseltamivir phosphate (OP) decreases the cleavage of sialic acid by Neu-1, thereby inhibiting receptor dimerization and downstream signaling. Cumulatively, the findings presented here indicate that targeting Neu-1 may inhibit the process of EMT and disrupt the metastatic potential of cancer cells. An ability to interfere with this process during the critical time period after the surgical removal of a cancerous tumor may offer a novel treatment strategy against a residual cancer cell population.

The limitations of this study are that we use three key cytokines based on the perioperative inflammatory and angiogenic cytokine profiles of surgical breast, colorectal, and prostate cancer patients and their clinical implications [11]. We reported [24] that cytokines can promote various cancer processes, such as angiogenesis, epithelial to mesenchymal transition (EMT), invasion, and tumor progression, and maintain cancer stem-cell-like (CSCs) cells. The mechanism(s) that continuously promote(s) tumors to progress in the TME still need(s) to be investigated.

## 6. Patents 

M.R.S. reports patents for using Neu-1 sialidase inhibitors in cancer treatment (Canadian patent no. 2858,246; United States patent no. US2015/0064282 A1; European patent no. 1874886.2; Chinese patent no. ZL201180076213.7; German patent no. 602011064575.7; Italian patent no. 502020000014650; UK patent no. 2773340; Swedish patent no. 2773340; Spanish patent no. 773340; Switzerland patent no. 2773340; and French patent no. 2773340). M.R.S. reports a patent for using oseltamivir phosphate and analogs thereof to treat cancer (International PCT patent no. PCT/CA2011/050690). W.W.H. and M.R.S. report a patent on a method to improve the effectiveness of anti-cancer therapies by exposing them to an inflammatory stimulus prior to treatment (Canadian patent no. PCT/CA2017/050765, pending). W.W.H. and M.R.S. report a patent on the compositions and methods for cancer treatment (Canadian patent no. PCT/CA2017/050768, pending). W.W.H. and M.R.S. report patents 55983477-9CN and CAN_DMS_150056368.1 on the COMPOSITIONS AND METHODS FOR THE TREATMENT OF CORONAVIRUS INFECTION AND RESPIRATORY COMPROMISE. 

## 7. Clinical Trials USFDA 

W.W.H. and M.R.S. have clinical trial approval to test OP in pancreatic cancer patients by the USFDA (clinical trial number #173874).

## Figures and Tables

**Figure 1 cancers-17-01234-f001:**
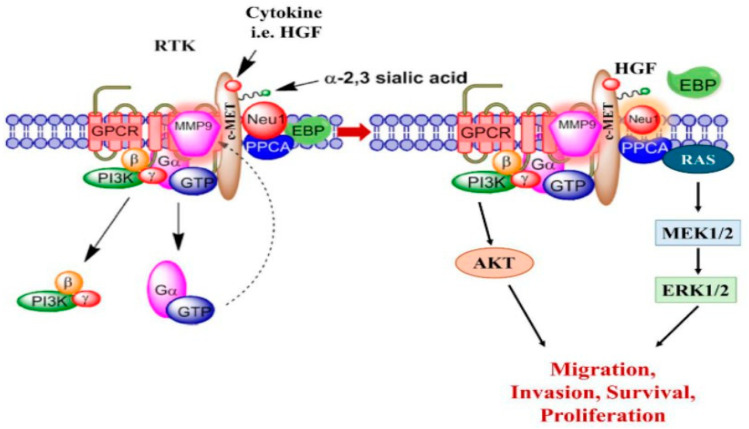
Cytokine receptor binding is implicated in the Neu-1-MMP-9-GPCR crosstalk. The binding of a cytokine to its receptor, a receptor tyrosine kinase (RTK), induces conformational changes in the associated GPCR, which in turn activates MMP-9 through the Gα subunit. This change activates Neu-1 by releasing the elastin-binding protein (EBP) from its complex with Neu-1 and protective protein cathepsin A (PPCA). Neu-1 cleaves α-2,3 sialic acid from the terminal glycosyl residue of the receptor, removing steric hindrance, which in turn induces the cytokine-dependent signal from the cytokine-binding receptor. Inhibitors of various components of this signaling paradigm are oseltamivir phosphate (OP; inhibitor of Neu-1), MMP9i (MMP-9 inhibitor), and BIM 23127 (NMBR inhibitor). Abbreviations: RTK: receptor tyrosine kinase; GPCR: G-protein-coupled receptor; MMP-9: matrix metalloprotease-9; Neu-1: neuraminidase-1; PPCA: protective protein cathepsin A; EBP: elastin-binding protein; HGF: hepatocyte growth factor; c-MET: HGF receptor. Citation. Modified in part from Research and Reports in Biochemistry 2013:3 17–30© 2013 Abdulkhalek et al., https://doi.org/10.2147/RRBC.S28430 (accessed on 7 January 2013) [29], publisher and licensee Dove Medical Press Ltd. This is an open-access article that permits unrestricted non-commercial use, provided the original work is properly cited.

**Figure 2 cancers-17-01234-f002:**
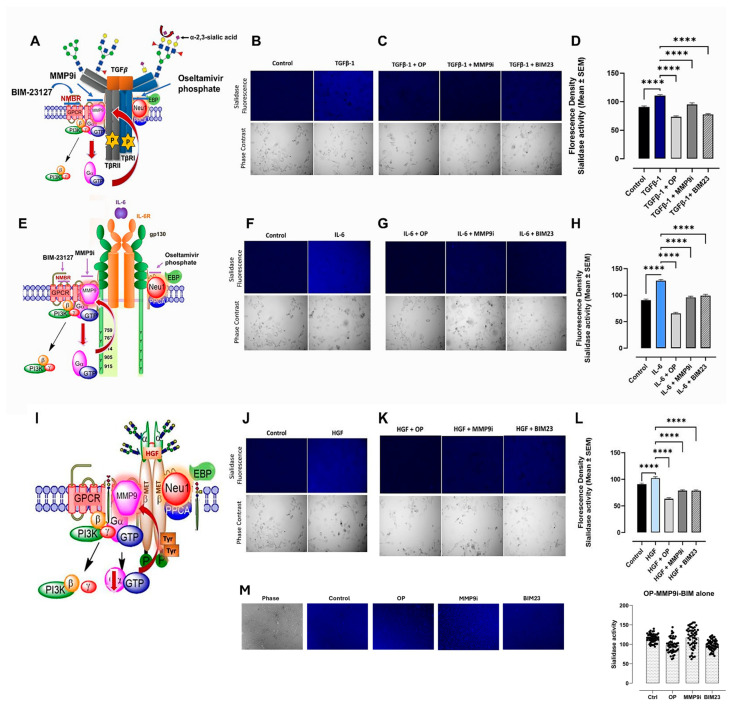
Cytokines increase sialidase activity in PANC-1 cells, while inhibitors of signaling paradigm OP, MMP9i, and BIM23 reverse cytokine activity. Pancreatic cancer cells (PANC-1) were treated with either TGFβ-1 (4.0 × 10^−3^ µg/mL) (**A**–**D**), IL-6 (4.1 × 10^−5^ µg/mL) (**E**–**H**), HGF (5.97 × 10^−4^ µg/mL) (**I**–**L**), and cells treated with inhibitors alone (**M**) for one minute. 4-MUNANA substrate was added to live cells to measure sialidase activity (emission 450 nm, excitation 365 nm). The addition of (**A**–**D**) TGFβ-1, (**E**–**H**) IL-6, and (**I**–**L**) HGF increased the sialidase fluorescence compared to the untreated control. In contrast, the use of inhibitors of Neu-1 (OP), MMP-9 (MMP9i), and NMBR/GPCR (BIM23) significantly decreased sialidase fluorescence in TGFβ-1 (**D**), IL-6 (**H**), and HGF (**L**) treatment groups. The mean ± SEM (biological replicates: 5, technical replicates: 50) fluorescence density of sialidase activity demonstrates a highly significant (*p* < 0.0001) downregulation of sialidase activity with all three inhibitors. Abbreviations: TGFβ-1: transforming growth factor beta-1; IL-6: interleukin-6; HGF: hepatocyte growth factor; OP: oseltamivir phosphate; MMP9i: MMP9 inhibitor; BIM23: BIM-23127. **** *p* < 0.0001.

**Figure 3 cancers-17-01234-f003:**
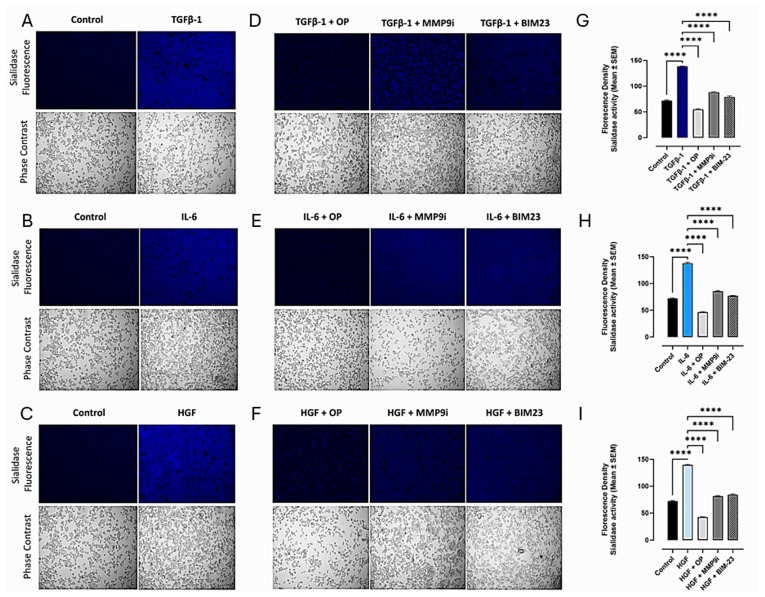
Cytokines increase sialidase activity in colorectal SW620 cells, while inhibitors of signaling paradigm OP, MMP9i, and BIM23 reverse cytokine activity. Colorectal cancer cells (SW620) were treated with either TGFβ-1 (4.0 × 10^−3^ µg/mL), IL-6 (4.1 × 10^−5^ µg/mL), or HGF (5.97 × 10^−4^ µg/mL). 4- MUNANA substrate was added to live cells to measure sialidase activity (emission 450 nm, excitation 365 nm). The addition of (**A**) TGFβ-1, (**B**) IL-6, and (**C**) HGF increased the sialidase fluorescence compared to the untreated control. In contrast, the use of inhibitors of Neu-1 (OP), MMP-9 (MMP9i), and NMBR/GPCR (BIM23) significantly decreased sialidase fluorescence in TGFβ-1, IL-6, and HGF treatment groups (**D**–**F**). The mean ± SEM (biological replicates: 5, technical replicates: 50) fluorescence density of sialidase activity (**G**–**I**) demonstrates a highly significant (*p* < 0.0001) downregulation of sialidase activity with all three inhibitors. Abbreviations: TGFβ-1: transforming growth factor beta-1; IL-6: interleukin-6; HGF: hepatocyte growth factor; OP: oseltamivir phosphate; MMP9i: MMP9 inhibitor; BIM23: BIM-23127. **** *p* < 0.0001.

**Figure 4 cancers-17-01234-f004:**
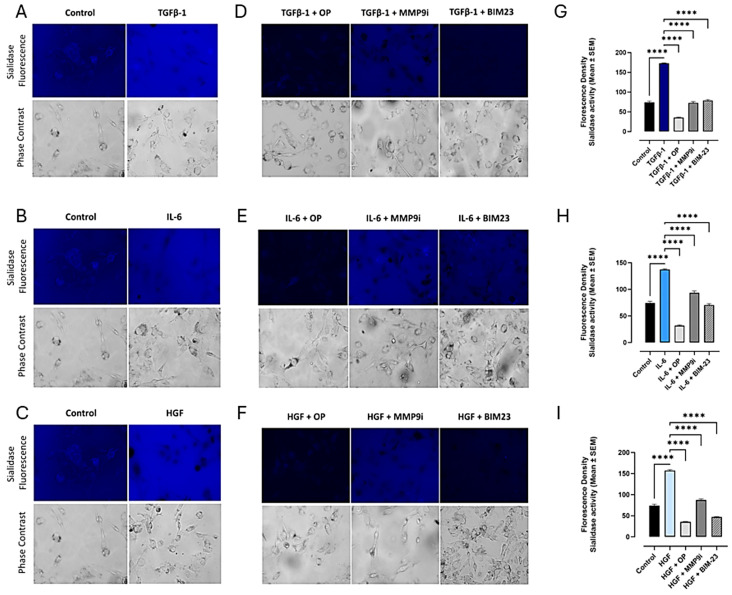
Cytokines increase sialidase activity in breast MCF-7 cells, while inhibitors of signaling paradigm OP, MMP9i, and BIM23 reverse cytokine activity. Breast cancer cells (MCF-7) were treated with either TGFβ-1 (4.0 × 10^−3^ µg/mL), IL-6 (4.1 × 10^−5^ µg/mL), or HGF (5.97 × 10^−4^ µg/mL). 4-MUNANA substrate was added to live cells to measure sialidase activity (emission 450 nm, excitation 365 nm). The addition of (**A**) TGFβ-1, (**B**) IL-6, and (**C**) HGF increased the sialidase fluorescence compared to the untreated control. In contrast, the use of inhibitors of Neu-1 (OP), MMP-9 (MMP9i), and NMBR/GCPR (BIM23) significantly decreased sialidase fluorescence in TGFβ-1, IL-6, and HGF treatment groups (**D**–**F**). The mean ± SEM (biological replicates: 5, technical replicates: 50) fluorescence density of sialidase activity (**G**–**I**) demonstrates a highly significant (*p* < 0.0001) downregulation of sialidase activity with all three inhibitors. Abbreviations: TGFβ-1: transforming growth factor beta-1; IL-6: interleukin-6; HGF: hepatocyte growth factor; OP: oseltamivir phosphate; MMP9i: MMP9 inhibitor; BIM23: BIM-23127. **** *p* < 0.0001.

**Figure 5 cancers-17-01234-f005:**
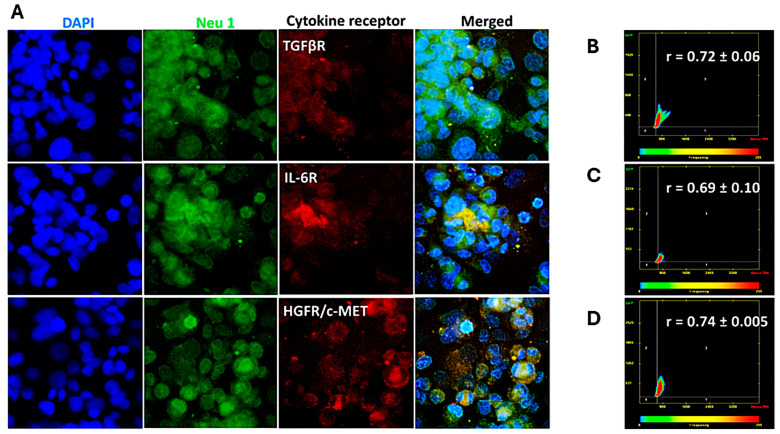
Receptors of TGFβ-1, IL-6, and HGF colocalize with neuraminidase-1 in PANC-1 cancer cells. Cells were fixed, permeabilized, blocked, and immunostained with the specific antibodies. PANC-1 cells were treated with Alexa Fluor™ 594-conjugated antibodies for the cytokine receptors TGFβR, IL-6R, and HGFR/c-MET and Alexa Fluor™ 488-conjugated Neu-1 antibody. (**A**) Merging of fluorescent images acquired from Zeiss M2 epi-fluorescent microscope (40× objective magnification) shows areas of colocalization through yellow fluorescence. Pearson correlation coefficient (R-value) measured a significant positive correlation (0.69 < r < 0.74) between the cytokine receptors (**B**) TGFβR, (**C**) IL-6R, and (**D**) HGFR/c-MET and Neu-1 (biological replicates: 4, technical replicates: 1). Abbreviations: TGFβR: transforming growth factor beta receptor; IL-6R: interleukin-6 receptor; HGFR/c-MET: hepatocyte growth factor receptor; Neu-1: neuraminidase-1.

**Figure 6 cancers-17-01234-f006:**
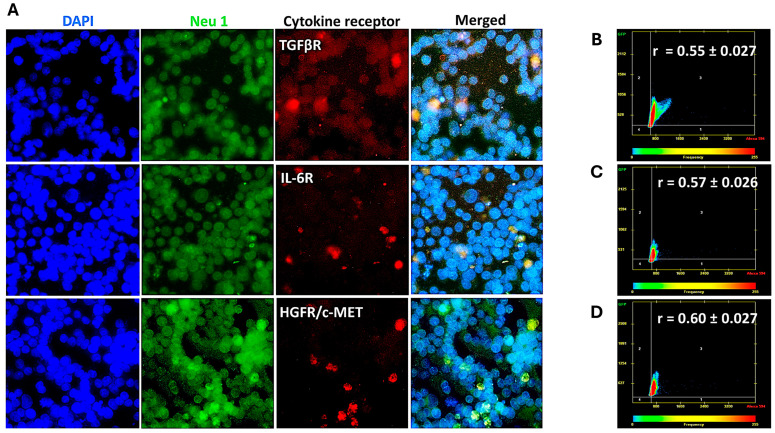
Receptors of TGFβ-1, IL-6, and HGF moderately colocalize with neuraminidase-1 in SW620 cancer cells. Cells were fixed, permeabilized, blocked, and immunostained with the specific antibodies. SW620 cells were treated with Alexa Fluor™ 594-conjugated antibodies for the cytokine receptors TGFβR, IL-6R, and HGFR/c-MET and Alexa Fluor™ 488-conjugated Neu-1 antibody. (**A**) Merging of fluorescent images acquired from Zeiss M2 epi-fluorescent microscope (40× objective magnification) shows areas of colocalization through yellow fluorescence. Pearson correlation coefficient (R-value) measured a moderate correlation (0.55 < r < 0.60) between the cytokine receptors (**B**) TGFβR, (**C**) IL-6R, and (**D**) HGFR/c-MET and Neu-1 (biological replicates: 4, technical replicates: 1). Abbreviations: TGFβR: transforming growth factor beta receptor; IL-6R: interleukin-6 receptor; HGFR/c-MET: hepatocyte growth factor receptor; Neu-1: neuraminidase-1.

**Figure 7 cancers-17-01234-f007:**
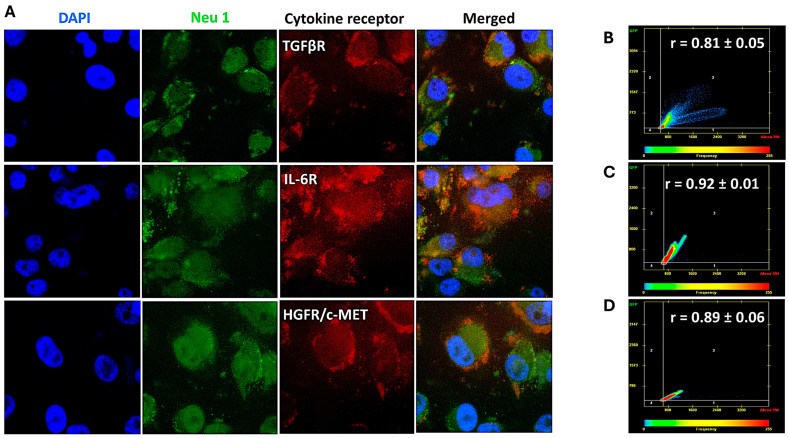
Receptors of TGFβ-1, IL-6, and HGF markedly colocalize with neuraminidase-1 in MCF-7 cancer cells. Cells were fixed, permeabilized, blocked, and immunostained with the specific antibodies. MCF-7 cells were treated with Alexa Fluor™ 594-conjugated antibodies for the cytokine receptors TGFβR, IL-6R, and HGFR/c-MET and Alexa Fluor™ 488-conjugated Neu-1 antibody. (**A**) Merging of fluorescent images acquired from Zeiss M2 epi-fluorescent microscope (40× objective magnification) shows areas of colocalization (yellow) fluorescence. Pearson correlation coefficient (R-value) measured a strong positive correlation (0.81 < r < 0.92) between the cytokine receptors (**B**) TGFβR, (**C**) IL-6R, and (**D**) HGFR/c-MET and Neu-1 (biological replicates: 4, technical replicates: 1). Abbreviations: TGFβR: transforming growth factor beta receptor; IL-6R: interleukin-6 receptor; HGFR/c-MET: hepatocyte growth factor receptor; Neu-1: neuraminidase-1.

**Figure 8 cancers-17-01234-f008:**
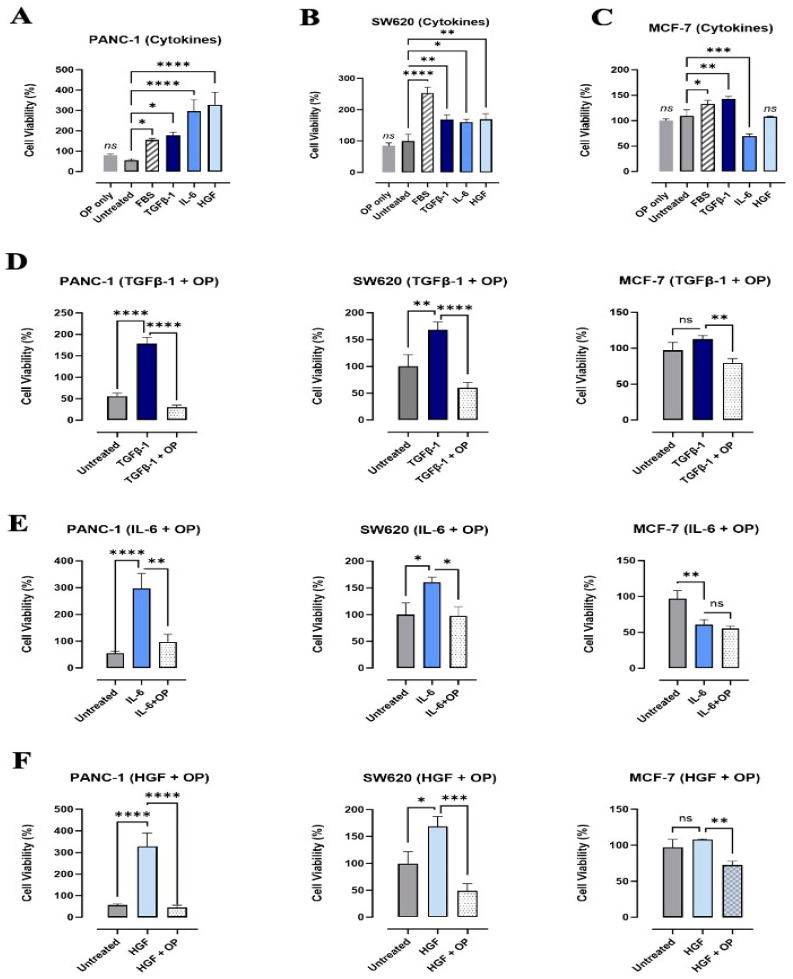
Cytokine treatment enhances the viability of cancer cells, whereas oseltamivir phosphate reduces viability in cytokine-treated cells but does not induce cell death. AlamarBlue cell viability assay tested changes in the viability of (**A**) PANC-1, (**B**) SW620, and (**C**) MCF-7 with the addition of cytokines and OP. In all three cell lines, FBS (medium) increased cell viability, while the addition of OP without any other factors did not impact cell viability compared to untreated control. In PANC-1 and SW620 cells, the addition of TGFβ-1, IL-6, and HGF significantly increased cell viability, whereas in MCF-7 cells, TGFβ-1 increased cell viability, IL-6 decreased cell viability, and HGF did not significantly impact cell viability. In all three cell lines, the combination of (**D**) TGFβ-1+ OP and (**F**) HGF + OP decreased cell viability compared to cytokine-treated cells. It brought the overall cell viability down to approximately untreated cell viability. IL-6 combination with OP (**E**) did not have an impact on cell viability in MCF-7; however, it did decrease cell viability to untreated levels in PANC-1 and SW620 cells. Statistical significance: * *p* < 0.0404; ** *p* < 0.0065; *** *p* < 0.0006; **** *p* < 0.0001 (biological replicates: 3, technical replicates: 12). Abbreviations: TGFβ-1: transforming growth factor beta-1; IL-6: interleukin-6; HGF: hepatocyte growth factor; OP: oseltamivir phosphate; FBS: fetal bovine serum.

**Figure 9 cancers-17-01234-f009:**
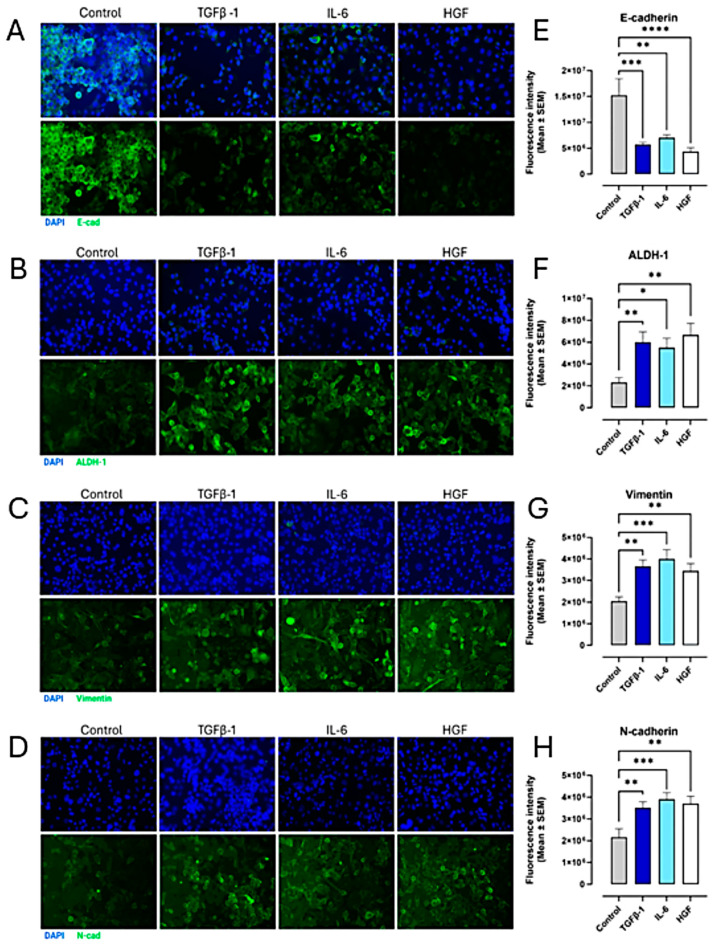
Cytokines upregulate the expression of mesenchymal markers and decrease the expression of epithelial markers in the immunofluorescence of pancreatic cancer cells (PANC-1). PANC-1 cells were treated with either TGFβ-1 (4.0 × 10^−3^ µg/mL), IL-6 (4.1 × 10^−5^ µg/mL), or HGF (5.97 × 10^−4^ µg/mL) for 24 h. Control cells were untreated. Cells were fixed, permeabilized, blocked, and immunostained with the specific antibodies for EMT markers. Fluorescent imaging of EMT cell markers (**A**) E-cadherin, (**B**) ALDH-1, (**C**) vimentin, and (**D**) N-cadherin stained with specific monoclonal antibody conjugated with Alexa Fluor 488^TM^ were acquired using Zeiss M2 epi-fluorescent microscope at 20× objective magnification. The nuclear stain used DAPI in the mounting medium, as shown in blue. Fluorescent intensity was measured and graphed for each maker (**E**–**H**). Each bar represents the mean with error bars denoting SEM (biological replicates: 8, technical replicates: 1). Statistical significance: * *p* < 0.0147; ** *p* < 0.0069; *** *p* <0.0008; **** *p* < 0.0001. Abbreviations: TGFβ-1: transforming growth factor beta-1; IL-6: interleukin-6; HGF: hepatocyte growth factor; E-cad: E-cadherin; N-cad: N-cadherin.

**Figure 10 cancers-17-01234-f010:**
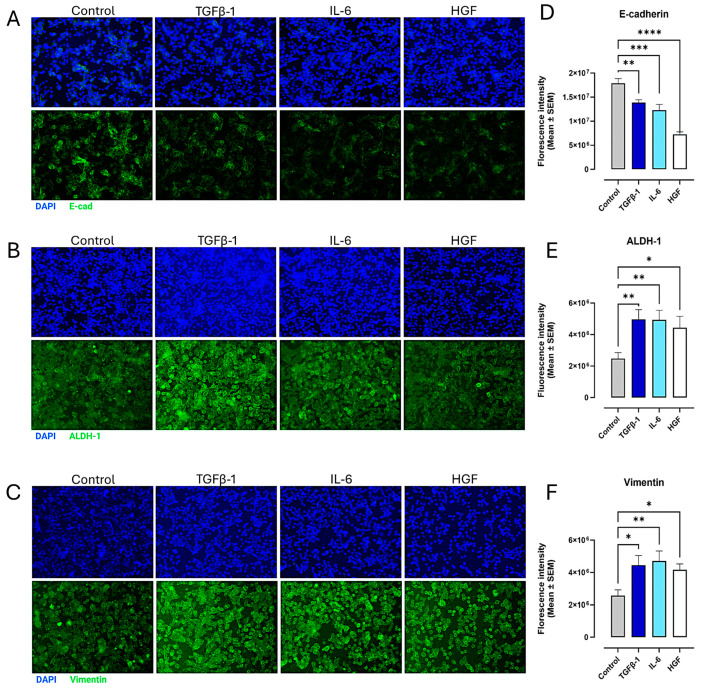
Cytokines upregulate the expression of mesenchymal markers and decrease the expression of epithelial markers in the immunofluorescence of colorectal cancer cells (SW620). SW620 cells were treated with either TGFβ-1 (4.0 × 10^−3^ µg/mL), IL-6 (4.1 × 10^−5^ µg/mL), or HGF (5.97 × 10^−4^ µg/mL) for 24 h. Control cells were untreated. Cells were fixed, permeabilized, blocked, and immunostained with the specific antibodies for EMT markers. Fluorescent imaging of EMT cell markers (**A**) E-cadherin, (**B**) ALDH-1, and (**C**) vimentin stained with specific antibody conjugated with Alexa Fluor™ 488 were acquired using Zeiss M2 epi-fluorescent microscope at 20× objective magnification. The nuclear staining used DAPI in the mounting medium, as shown in blue. Fluorescent intensity was measured and graphed for each maker (**D**–**F**). Each bar represents the mean with error bars denoting SEM (biological replicates: 8, technical replicates: 1). Statistical significance: * *p* < 0.028; ** *p* < 0.0071; *** *p* = 0.0001; **** *p* < 0.0001. Abbreviations: TGFβ-1: transforming growth factor beta-1; IL-6: interleukin-6; HGF: hepatocyte growth factor; E-cad: E-cadherin; N-cad: N-cadherin.

**Figure 11 cancers-17-01234-f011:**
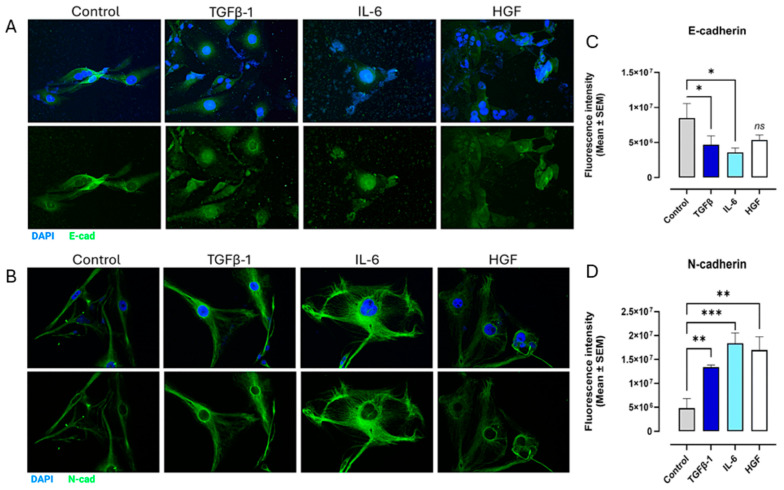
Cytokines upregulate the expression of mesenchymal marker N-cadherin and decrease epithelial marker E-cadherin in breast cancer cells (MCF-7). MCF-7 cells were treated with either TGFβ-1 (4.0 × 10^−3^ µg/mL), IL-6 (4.1 × 10^−5^ µg/mL), or HGF (5.97 × 10^−4^ µg/mL) for 24 h. Control cells were untreated. Cells were fixed, permeabilized, blocked, and immunostained with the specific antibodies for EMT markers. Fluorescent imaging of EMT cell markers (**A**) E-cadherin and (**B**) N-cadherin stained with specific antibody conjugated with Alexa Fluor™ 488 were acquired using Zeiss M2 epi-fluorescent microscope at 20× objective magnification. The nuclear staining used DAPI, as shown in blue. (**C**,**D**) Fluorescent intensity was measured and graphed for each maker. Each bar represents the mean with error bars denoting SEM (biological replicates: 8, technical replicates: 1). Statistical significance: * *p* < 0.045; ** *p* < 0.0059; *** *p* = 0.0003. Abbreviations: TGFβ-1: transforming growth factor beta-1; IL-6: interleukin-6; HGF: hepatocyte growth factor; E-cad: E-cadherin; N-cad: N-cadherin; ns: not significant.

**Figure 12 cancers-17-01234-f012:**
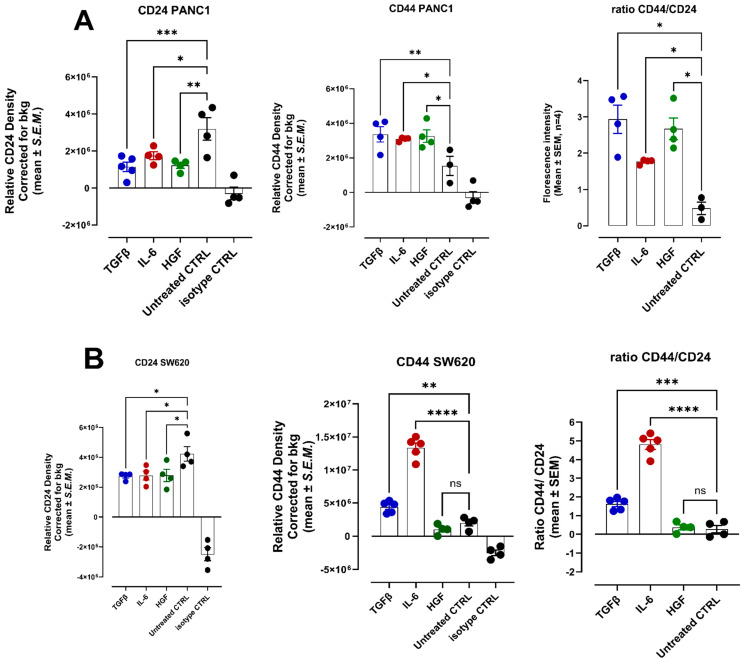
TGFβ-1, IL-6, and HGF induce CD44 and CD24 as markers for identifying and characterizing cancer stem cells (CSCs) in PANC-1 and SW620 cancer cells. Expression of CD44, CD24, and ratio CD44/24 on (**A**) PANC-1 and (**B**) SW620 cells following treatment with TGFβ-1 (4.0 × 10^−3^ µg/mL), IL-6 (4.1 × 10^−5^ µg/mL), or HGF (5.97 × 10^−4^ µg/mL) for 24 h. Relative density is reported as a mean ± SEM of 5 images of stained cells per treatment. Fluorescent imaging of EMT cell markers CD44 and CD24 stained with specific antibodies conjugated with Alexa Fluor™ 488 was acquired using a Zeiss M2 epi-fluorescent microscope at 20× objective magnification. Each bar represents the mean with error bars denoting SEM (biological replicates: 5, technical replicates: 1). Statistical significance: * *p* < 0.045; ** *p* < 0.0059; *** *p* = 0.0003; **** *p* < 0.0001; and ns = non-significance. Abbreviations: TGFβ-1: transforming growth factor beta-1; IL-6: interleukin-6; HGF: hepatocyte growth factor.

**Figure 13 cancers-17-01234-f013:**
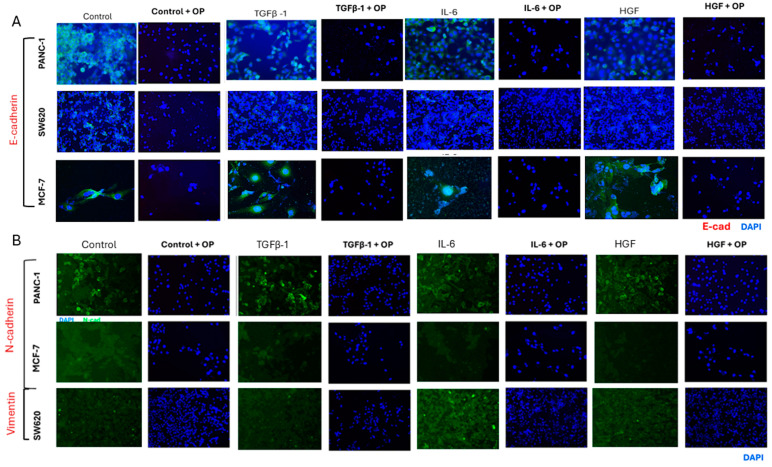
Immunofluorescence with the addition of OP does not show expression of any EMT markers. An immunofluorescence assay was conducted to determine the impact of OP on markers of EMT. The same protocol was used with the change in an Alexa Fluor 488™ conjugated antibody for EMT markers. Cells (200,000 cells/well) were treated with OP for 15 min. Cells were fixed, permeabilized, blocked, and immunostained with the specific antibodies for EMT markers. Fluorescent imaging of EMT cell markers E-cadherin, vimentin, and N-cadherin stained with Alexa Fluor™ 488 conjugated antibody was acquired using Zeiss M2 epi-fluorescent microscope at 20× objective magnification. (**A**) No fluorescence was noted for E-cadherin following OP treatment in the three cell lines compared to the cytokine-treated controls. (**B**) Similar findings were seen with mesenchymal markers N-cadherin and vimentin. DAPI nuclear staining shows an intact nucleus, indicating that the cells are intact; they do not express cell surface markers for EMT (biological replicates: 3, technical replicates: 1). Abbreviations: TGFβ-1: transforming growth factor beta-1; IL-6: interleukin-6; HGF: hepatocyte growth factor; OP: oseltamivir phosphate; E-cad: E-cadherin; N-cad: N-cadherin.

**Figure 14 cancers-17-01234-f014:**
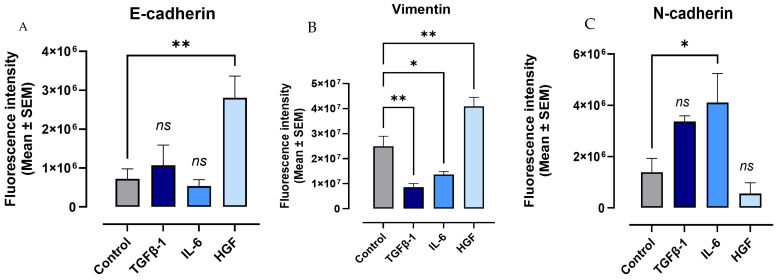
Immunofluorescence for EMT markers in RAW-Blue macrophages. RAW-Blue macrophages were treated with cytokines following the same protocol as in Figure 8, Figure 9 and Figure 10. Fluorescent imaging of EMT cell surface markers (**A**) E-cadherin, (**B**) vimentin, and (**C**) N-cadherin stained with Alexa Fluor™ 488 were acquired using Zeiss M2 epi-fluorescent microscope at 20× objective magnification. The findings show contrasting findings to those seen in the cancer cells (PANC-1, SW620, and MCF-7). The epithelial marker E-cadherin (**A**) demonstrated a significant increase (** *p* = 0.0038) in HGF-treated cells, while TGFβ-1 and IL-6 were insignificant. Mesenchymal vimentin (**B**) demonstrated a downregulated expression in TGFβ (** *p* = 0.0012) and IL-6 (* *p* = 0.0122) and an increased expression in HGF (*p* = 0.0024)-treated macrophages. The other mesenchymal marker, N-cadherin (**C**), also had significantly upregulated levels in IL-6 (* *p* = 0.0149), while TGFβ-1, even though it is higher than untreated cells, was insignificantly higher. HGF-treated cells saw a decreased expression of N-cadherin, although it was also insignificant. Data are represented as mean ± SEM (biological replicates: 4, technical replicates: 1). Abbreviations: TGFβ-1: transforming growth factor beta-1; IL-6: interleukin-6; HGF: hepatocyte growth factor; ns: not significant.

## Data Availability

All data needed to evaluate this paper’s conclusions are present. The preclinical data sets generated and analyzed during the current study are not publicly available but are available from the corresponding author upon reasonable request. The data will be provided following the review and approval of a research proposal, statistical analysis plan, and execution of a data-sharing agreement. The data will be accessible for twelve months for approved requests, considering possible extensions; contact szewczuk@queensu.ca for more information on the process or to submit a request.

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
