# Peer review of "Pro-Inflammatory Cytokines Transactivate Glycosylated Cytokine Receptors on Cancer Cells to Induce Epithelial–Mesenchymal Transition to the Metastatic Phenotype"

_cancers, 2025, doi:10.3390/cancers17071234_

Round 1
Reviewer 1 Report
Comments and Suggestions for Authors
In this article, the authors hypothesized that the binding of TGF-β1, IL-6, and HGF cytokines to their respective receptors induces cancer cells to adopt a mesenchymal phenotypic property through the Neu-1-MMP9-GPCR crosstalk signaling platform. In my opinion, the manuscript requires major improvements:
- Figures 2, 3 and 4. The panels are too small.
- Cytokine Receptors Colocalize with Neu1 on the Cell Surface of Naïve Unstimulated PANC- 332, SW-620 and MCF7 Cells. A validation by immunoprecipitation is suggested.
- Line 388. TGFb-1, IL-6 and HGF induce the EMT phenotype in PANC-1, SW620 and MCF-7 cancer cells. The EMT induction should be better investigated. qPCR and western blotting validations of a larger set of markers is necessary.
- Figure 10. MCF-7 cells have a cobblestone morphology and not this elongated shape. These cells do not express N-cadherin.
Author Response
REVIEWER #1
In this article, the authors hypothesized that the binding of TGF-β1, IL-6, and HGF cytokines to their respective receptors induces cancer cells to adopt a mesenchymal phenotypic property through the Neu-1-MMP9-GPCR crosstalk signaling platform. In my opinion, the manuscript requires major improvements:
Figures 2, 3 and 4. The panels are too small.
Cytokine Receptors Colocalize with Neu1 on the Cell Surface of Naïve Unstimulated PANC- 332, SW-620 and MCF7 Cells. A validation by immunoprecipitation is suggested.
Author response: We have enlarged the figures 2-4. They have a resolution of 4000x2250 pixels (2.2 MB). The colocalization experiments were done to just confirm the results of the sialidase experiments for each of the cytokine receptors using specific inhibitors of the signaling paradigm. We have published peer-reviewed articles in the same Cells journal using colocalization experiments to confirm the sialidase results with CB2 cannabinoids GPCR heterodimerization with glycosylated receptors (Cells 2024, 13, 480. https://doi.org/10.3390/cells13060480) without coimmunoprecipitation.
In addition, we have published peer-reviewed articles confirming the crosstalk of TLR-7 with Neu1 using coimmunoprecipitation experiments: IP: TLR7 Blot: Neu1 and IP:Neu1 Blot TLR7 (http://dx.doi.org/10.1016/j.cellsig.2013.06.010). Also, Neuromedin B receptor (NMBR) co-immunoprecipitates with TLR4 (http://dx.doi.org/10.1016/j.cellsig.2012.06.016). Neu1 co-immunoprecipitates with insulin receptors IRβ (http://dx.doi.org/10.1016/j.cellsig.2014.02.015).
Line 388. TGFb-1, IL-6 and HGF induce the EMT phenotype in PANC-1, SW620 and MCF-7 cancer cells. The EMT induction should be better investigated. qPCR and western blotting validations of a larger set of markers is necessary.
Author response: qPCR can be used to identify genes that are upregulated or downregulated during EMT, providing insights into the molecular mechanisms underlying this process. qPCR, or quantitative polymerase chain reaction, is a powerful technique used to measure the expression levels of genes. We recently published a review article highlighting on how the environmental impact of cannabinoid factors and the subsequent combined effects on epigenetic modifications, such as cancer metabolism, through a biased G protein‐coupled receptor signaling paradigm regulating several hallmarks of cancer (Cancers 2023, 15, 1030. https://doi.org/10.3390/cancers15041030). Significant epigenetic changes detected are DNA methylation, histone modifications, and RNA‐associated alterations, all of which can affect qPCR results. Cancer epigenetics have shown extensive reprogramming of every component of the epigenetic machinery in cancer development, including DNA methylation, histone modifications, nucleosome positioning, non‐coding RNAs, and microRNA expression. The review focuses on the role of cannabinoid receptors CB1 and CB2 signaling in activating numerous receptor tyrosine kinases and Toll‐like receptors in the induction of epigenetic landscape alterations in cancer cells, which might transmogrify cancer metabolism and epigenetic reprogramming to a metastatic phenotype.
We are in the process of investigating the epigenetic reprogramming to a metastatic phenotype, which is an ongoing research program.
Figure 10. MCF-7 cells have a cobblestone morphology and not this elongated shape. These cells do not express N-cadherin.
Author response: TGF-β1: TGF-β1 can induce EMT in MCF-7 cells, marked by an increase in mesenchymal markers (N-cadherin and Vimentin) and a loss of epithelial markers (E-cadherin) (Cancer Cell Int 19, 343 (2019). https://doi.org/10.1186/s12935-019-1068-7). The elongated shape of the MCF-7 cells reveal an "aggressive" MCF-7 breast cancer cells (Breast Cancer Res Treat 2014 Nov;148(2):269-77. doi: 10.1007/s10549-014-3159-4), which would express the N-cadherin markers.
Also, the lymph node metastasis (SW620) cells, are motile cells having an elongated form with pseudopodia whereas compared with non-motile colon carcinoma (SW480) cells are characterized by a round shape.
Overall Conclusions: We have added our patents granted and most recent an official approval from USFDA to test OP in pancreatic cancer patients (Clinical trial number #173874).
Submission Date
26 February 2025
Date of this review
07 Mar 2025 13:25:02
Reviewer 2 Report
Comments and Suggestions for Authors
Comments:
- The study demonstrates cytokine-induced sialidase activity but lacks mechanistic insight into how Neu-1 activation influences downstream signaling beyond receptor activation. It is advisable for the authors to measure phosphorylation levels of key RTKs (e.g., c-MET, TGFβR, IL-6R) upon cytokine treatment, with and without OP, to confirm Neu-1's regulatory role. Using siRNA or CRISPR knockout of Neu-1 would further validate whether Neu-1 is necessary for cytokine-induced receptor activation.
- In alignment with previous comments, it is intriguing if author validate whether Neu-1 is necessary for cytokine-induced receptor activation using existing technologies such as Neu-1 specific siRNA or CRISPR knockout.
- The study does not establish direct interactions between Neu-1 and the cytokine receptors. To further confirm these interactions, co-immunoprecipitation (Co-IP) or proximity ligation assays (PLA) should be employed.
- How quickly Neu-1 is activated upon cytokine stimulation? A time-course experiment should be conducted to conclude the findings.
- Author observed strong correlation of cell membrane colocalization between cytokine receptors with Neu-1 in PANC-1 and MCF-7 but only a moderate in SW620 cells. Does this imply cell-type-specific differences. If so, further investigation is needed to explore whether SW620 cells rely on alternative signaling pathways for cytokine-induced responses.
- To understand metastatic plasticity, reversibility of cytokine-induced EMT is crucial, which is not addressed by authors experimentally. It is advisable if authors should address this mechanism under specific conditions.
- Differences in EMT markers among cell lines (e.g., SW620 lacking N-cadherin, MCF-7 lacking vimentin) suggest that each cell line follows a distinct EMT program. To rule out discrepancies, the authors should perform mesenchymal-to-epithelial transition (MET) assays by removing cytokines and tracking marker expression over time or use single-cell RNA sequencing to identify distinct EMT states and heterogeneity within treated populations.
- OP completely blocks EMT markers following cytokine treatment. However, the study does not address whether OP affects cell viability. To clarify, cell viability and proliferation assays should be conducted to ensure that OP’s effects are specific to EMT inhibition and not due to general cytotoxicity.
- In alignment with previous comments, migration/invasion assays should also be performed to assess the functional consequences of EMT inhibition.
- Cytokine-treated macrophages data indicates hybrid EMT state which contradicts with the clear EMT transition seen in cancer cells. The authors should either perform RNA-seq analysis to compare transcriptional profiles of macrophages and cancer cells undergoing cytokine-induced EMT or investigate whether the hybrid EMT state in macrophages enhances their immunosuppressive function or interaction with cancer cells.
- Despite only moderate colocalization of Neu-1 with cytokine receptors in SW620 cells, strong EMT changes are observed. This suggests that Neu-1 may not be the only driver of cytokine-induced EMT. The authors should test alternative pathways (e.g., STAT3, SMAD-dependent pathways) in SW620 cells to further elucidate the mechanisms involved.
- The study lacks appropriate controls for non-cytokine-related Neu-1 activation. It is unclear whether Neu-1 activity is exclusive to cytokine stimulation or if other pathways contribute. The authors should use non-cytokine stimuli (e.g., serum stimulation, growth factors) to determine whether Neu-1 activation is specific to cytokine-induced signaling.
- The use of OP to inhibit Neu-1 is well-executed, but genetic validation (e.g., Neu-1 knockout) is missing. CRISPR/Cas9 knockout of Neu-1 would confirm that its inhibition, and not off-target effects of OP, blocks EMT.
- The impact of cytokine concentration and treatment time is mentioned but not systematically analyzed. No dose-response experiments are provided. The authors should conduct dose-response studies to determine the optimal cytokine concentration and treatment time for inducing EMT.
- The hypothesis regarding E-cadherin cleavage (membrane vs. soluble) is mentioned but not experimentally validated. The authors should perform Western blotting for soluble versus membrane-bound E-cadherin to confirm this hypothesis.
- Contradictory findings in macrophages (e.g., increased E-cadherin and vimentin with HGF, mixed results for IL-6) are discussed but lack mechanistic validation. The authors should investigate the underlying mechanisms behind these contradictory results to better understand the role of cytokines in macrophage plasticity.
- The claim that cytokines post-surgical resection drive metastasis is not directly tested. There is no in vivo validation using post-surgical models. The authors should incorporate in vivo studies to validate their claims about cytokine-driven metastasis after surgery.
- The translation of Neu-1 inhibition to therapy lacks preclinical evidence. No data are provided to show that OP reduces metastasis in vivo. Preclinical models should be used to validate whether OP can reduce metastasis.
- The conclusion claims that OP induces senescence, but this is inferred from reduced wound healing and tunneling nanotube formation. The authors should conduct experiments involving SA-β-Gal staining and analyze p16/p21 expression following OP treatment to confirm senescence instead of apoptosis or differentiation.
- OP has been linked to Neu-1 inhibition, but its anti-metastatic role is not widely established. The authors should further validate the anti-metastatic potential of OP in vivo.
- The representative micrographs and immunofluorescence images are too small to accurately correlate the results. The authors are advised to provide higher resolution and magnified images, especially in Figures 2–4, to avoid discrepancies and improve image quality.
Author Response
REVIEWER #2
Comments:
The study demonstrates cytokine-induced sialidase activity but lacks mechanistic insight into how Neu-1 activation influences downstream signaling beyond receptor activation. It is advisable for the authors to measure phosphorylation levels of key RTKs (e.g., c-MET, TGFβR, IL-6R) upon cytokine treatment, with and without OP, to confirm Neu-1's regulatory role. Using siRNA or CRISPR knockout of Neu-1 would further validate whether Neu-1 is necessary for cytokine-induced receptor activation.
Author response: In 2009, we published an article using primary BM macrophages were derived from wild-type mice (WT), hypomorphic cathepsin A mice with the secondary ~90% reduction of the Neu1 activity (Neu1-CathA KD), and ce with normal Neu1 sialidase bound to inactive cathepsin A Ser190Alamutant (Neu1-CathA KI). Activation of Neu1 is induced by TLR ligands binding to their respective receptors. Here, we showed that endotoxin lipopolysaccharide (LPS)-induced MyD88/TLR4 complex formation and subsequent NFκB activation is dependent on the removal of α-2,3-sialyl residue linked to β-galactoside of TLR4 by theNeu1 activity associated with LPS-stimulated live primary macrophage cells, macrophage and dendritic cell lines but not with primary Neu1-deficient macrophage cells. These same TLR ligand-induced NFκB responses are not observed in TLR deficient HEK293 cells, but are re-established in HEK293 cells stably transfected with TLR4/MD2, and are significantly inhibited by α-2,3-sialyl specific Maackia amurensis (MAL-2) lectin, α-2,3-sialyl specific galectin-1 and neuraminidase inhibitor Tamiflu but not by α-2,6-sialyl specific Sambucus nigra lectin (SNA). Taken together, the findings suggest that Neu1 desialylation of α-2,3-sialyl residues of TLR receptors enables in removing a steric hinderance to receptor association for TLR activation and cellular signaling.
In alignment with previous comments, it is intriguing if author validate whether Neu-1 is necessary for cytokine-induced receptor activation using existing technologies such as Neu-1 specific siRNA or CRISPR knockout.
Author response: Please see previously above response to this statement.
The study does not establish direct interactions between Neu-1 and the cytokine receptors. To further confirm these interactions, co-immunoprecipitation (Co-IP) or proximity ligation assays (PLA) should be employed.
Author response: Please see previously above response to this statement.
How quickly Neu-1 is activated upon cytokine stimulation? A time-course experiment should be conducted to conclude the findings.
Author response: The max Neu1 activity using live cells is within 1 minute. We have indicated this in the revised manuscript. We have published the protocol of the sialidase assay: Amith, Jayanth et al. (2010) JoVE_Protocol_2142 (DOI 10.37912142 )
Author observed strong correlation of cell membrane colocalization between cytokine receptors with Neu-1 in PANC-1 and MCF-7 but only a moderate in SW620 cells. Does this imply cell-type-specific differences. If so, further investigation is needed to explore whether SW620 cells rely on alternative signaling pathways for cytokine-induced responses.
Author response: A recent report revealed that two human cell lines, one established from a colon carcinoma (SW480) and the other from its lymph node metastasis (SW620), were compared with respect to their migration capacity employing a three-dimensional collagen matrix and time-lapse video recording. Non-motile cells were characterized by a round shape, whereas motile cells appeared in an elongated form with pseudopodia (https://doi.org/10.1016/S0304-3835(98)00201-8).
The SW620 cells demonstrated a moderate positive correlation (TGFβ-1R: r = 0.55 ± 0.00; IL-6R: r ₌ 0.57 ± 0.03; HGFR/c-MET: r ₌ 0.60 ± 0.00) (Figure 6). Interestingly, Matsuo et al. [25] ( Journal of International Medical Research 2003; 31: 69 – 75) demonstrated that SW620 cells have a higher level of IL-6 receptor secretion than the other colorectal cancer cell lines. Noteworthy, the concentrations IL-6R in each cell supernatant were measured by enzyme-linked immunosorbent assay on the third day of culture. In the present study, the SW620 cells were adhered to the assay plates for 24 hrs and assayed for colocalization.
To understand metastatic plasticity, reversibility of cytokine-induced EMT is crucial, which is not addressed by authors experimentally. It is advisable if authors should address this mechanism under specific conditions.
Author response: This is an interesting and important statement. This question focuses on the future design of experiments. In particular, we are focused on analyzing the epigenetic rewiring by these cytokines through the signaling paradigm noted in Figure 1. It is noteworthy that in Figure 12, Immunofluorescence with the addition of OP does not show expression of any EMT markers. An immunofluorescence assay was done to determine the impact of OP on markers of EMT.
Differences in EMT markers among cell lines (e.g., SW620 lacking N-cadherin, MCF-7 lacking vimentin) suggest that each cell line follows a distinct EMT program. To rule out discrepancies, the authors should perform mesenchymal-to-epithelial transition (MET) assays by removing cytokines and tracking marker expression over time or use single-cell RNA sequencing to identify distinct EMT states and heterogeneity within treated populations.
Author response: It is noteworthy that in Figure 12, Immunofluorescence with the addition of OP does not show expression of any EMT markers. An immunofluorescence assay was done to determine the impact of OP on markers of EMT. We have recently reported (Cancers 2022, 14, 3595. https://doi.org/10.3390/cancers14153595) that neuraminidase-1 (Neu-1) regulates the activation of several receptor tyrosine kinases implicated in EMT induction, angiogenesis, and cellular proliferation. In the report, continuous therapeutic targeting of Neu-1 using parenteral perfusion of oseltamivir phosphate (OP) and aspirin (ASA) with gemcitabine (GEM) treatment significantly disrupts tumor progression, critical compensatory signaling mechanisms, EMT program, CSC, and metastases in a preclinical mouse model of human pancreatic cancer. ASA- and OP-treated xenotumors significantly inhibited the metastatic potential when transferred into animals.
OP completely blocks EMT markers following cytokine treatment. However, the study does not address whether OP affects cell viability. To clarify, cell viability and proliferation assays should be conducted to ensure that OP’s effects are specific to EMT inhibition and not due to general cytotoxicity.
Author response: Thank you for this comment. We have done cell viability assays to examine the effectiveness of OP not due to cytotoxicity. We have included a new Figure in that cCytokine treatment enhances the viability of cancer cells, whereas oseltamivir phosphate reduces viability in cytokine-treated cells but does not induce cell death.
In alignment with previous comments, migration/invasion assays should also be performed to assess the functional consequences of EMT inhibition.
Author response: Thank you for this comment. We have extensive data on the very topic of migration/invasion assays with these cytokine. These data are part of another manuscript in preparation for submission for publication.
Cytokine-treated macrophages data indicates hybrid EMT state which contradicts with the clear EMT transition seen in cancer cells. The authors should either perform RNA-seq analysis to compare transcriptional profiles of macrophages and cancer cells undergoing cytokine-induced EMT or investigate whether the hybrid EMT state in macrophages enhances their immunosuppressive function or interaction with cancer cells.
Author response: Thank you for this comment. We have added that following statement: Hybrid EMT refers to a state where cancer cells retain some epithelial characteristics while simultaneously acquiring mesenchymal features, rather than undergoing a complete transition to a fully mesenchymal state [40].
Despite only moderate colocalization of Neu-1 with cytokine receptors in SW620 cells, strong EMT changes are observed. This suggests that Neu-1 may not be the only driver of cytokine-induced EMT. The authors should test alternative pathways (e.g., STAT3, SMAD-dependent pathways) in SW620 cells to further elucidate the mechanisms involved.
Author response: Thank you for this comment. We have reported preclinical animal studies on this very topic (A.M. Gilmour et al. / Cellular Signalling 25 (2013) 2587–2603). To further confirm the in vivo efficacy of OP therapy, the profiles of multiple phosphorylated proteins in the tumor lysates involved in cell signaling pathways were investigated using Western blot analyses and the Bio-Plex multiplex format to detect the profiles of multiple phosphorylated proteins in a single tumor sample lysate. This latter assay protocol is based on 6.5 μm magnetic beads which are optimized for high sensitivity and higher specificity in minimizing cross-reactivity with broad dynamic range using highly specific antibodies. Individual tumors taken from the OP treated cohorts expressed a significant less phosphorylation of EGFR-Tyr1173, Stat1-Tyr701, and NFκBp65-Ser311 compared to the untreated cohort as determined by Western blot analyses. The Bio-Plex multiplex format also showed a reduction in phosphorylation of Akt-Thr308, PDGFRα-Tyr754 and STAT1-Tyr701, but unexpectedly, an increased in phospho-Smad2-Ser465/467 and phospho-VEGFR2-Tyr1175 in the tumor lysates from the Tamiflu treated cohort compared to the untreated cohort. In contrast, OP treatment increased the phospho-protein end-points of SMAD2-Ser465/467 and VEGFR2-Tyr1175 compared to the untreated cohort.
The study lacks appropriate controls for non-cytokine-related Neu-1 activation. It is unclear whether Neu-1 activity is exclusive to cytokine stimulation or if other pathways contribute. The authors should use non-cytokine stimuli (e.g., serum stimulation, growth factors) to determine whether Neu-1 activation is specific to cytokine-induced signaling.
Author response: Thank you for this comment. We have addressed this question in the new Figure 8.
The use of OP to inhibit Neu-1 is well-executed, but genetic validation (e.g., Neu-1 knockout) is missing. CRISPR/Cas9 knockout of Neu-1 would confirm that its inhibition, and not off-target effects of OP, blocks EMT.
Author response: We have reported on the sialidase activity using primary macrophages obtained from Neu1-deficient mice (http://dx.doi.org/10.1016/j.cellsig.2012.06.016). To confirm that Neu1 activity, we used primary bone marrow macrophages derived from the hypomorphic cathepsin
A mice with secondary ∼90% reduction of the Neu1 activity (Neu1–CathA KD) [Circulation 117 (15) (2008) 1973–1981]. For controls, we used primary macrophages derived from wild-type (WT) mice and from cathepsinA KD mice (CathA KD, normal Neu1 sialidase bound to inactive cathepsin A S190A point mutation). After 7 days in culture with conditioned medium containing monocyte colony-stimulating factor, the live primary macrophage cells were treated with either the TLR4 ligand LPS, bombesin, the TLR3 ligand poly-IC or the TLR2 ligand killed M. butyricum (MYCO) to induce sialidase activity. The data indicated that neither LPS, MYCO nor bombesin induced sialidase activity in the live primary macrophage cells derived from these Neu1-deficient mice, but that they did in cells from the WT and CathA KD mice. These latter data provide additional evidence for Neu1 involvement in the bombesin-induced NMBR activation of Neu1 sialidase activity. We are supporting evidence from our previously peer-reviewed publications that Neu-1 is involved in the signaling paradigm as depicted in Figure 1.
Furthermore, we have reported ( Glycoconj J (2009) 26:1197–1212 DOI 10.1007/s10719-009-9239-8 ) on the central concept to this process. Neu1 and not Neu2,-3 and-4 form a complex with TLR-2,-3 and-4 on the cell surface of naïve macrophage cells. Neuraminidase inhibitors BCX1827, 2-deoxy-2,3-dehydro-N-acetylneuraminic acid (DANA), zanamivir and oseltamivir carboxylate have a limited significant inhibition of the LPS-induced sialidase activity in live BMC-2 macrophage cells but oseltamivir phosphate (OP) completely blocked this activity. OP inhibits LPS-induced sialidase activity in live BMC-2 cells with an IC50 of 1.2μM compared to an IC50 of 1015μM for its hydrolytic metabolite oseltamivir carboxylate. OP blockage of LPS-induced Neu1 sialidase activity is not affected in BMC-2 cells pretreated with anti-carboxylesterase agent clopidogrel. Endotoxin LPS binding to TLR4 induces Neu1 with subsequent activation of NFκB and the production of nitric oxide and proinflammatory IL-6 and TNFα cytokines in primary and macrophage cell lines. Hypomorphic cathepsin A mice with a secondary Neu1 deficiency responded poorly to LPS induced pro-inflammatory cytokines compared to the wild-type or hypomorphic cathepsin A with normal Neu1 mice. Our findings establish an unprecedented mechanism for pathogen molecule-induced TLR activation and cell function, which is critically dependent on Neu1 sialidase activity associated with TLR ligand treated live primary macrophage cells and macrophage and dendritic cell lines.
The impact of cytokine concentration and treatment time is mentioned but not systematically analyzed. No dose-response experiments are provided. The authors should conduct dose-response studies to determine the optimal cytokine concentration and treatment time for inducing EMT.
Author response: Thank you for this comment. The main focus of the study is to investigate the cytokine concentrations that spiked following surgical resections of tumors in patients (Cells 2023, 12, 2767.https://doi.org/10.3390/cells12232767). Noteworthy, the effectiveness of the concentrations used in this study showed proof of evidence for inducing EMT.
The hypothesis regarding E-cadherin cleavage (membrane vs. soluble) is mentioned but not experimentally validated. The authors should perform Western blotting for soluble versus membrane-bound E-cadherin to confirm this hypothesis.
Author response: We have removed this statement from the revised manuscript.
Contradictory findings in macrophages (e.g., increased E-cadherin and vimentin with HGF, mixed results for IL-6) are discussed but lack mechanistic validation. The authors should investigate the underlying mechanisms behind these contradictory results to better understand the role of cytokines in macrophage plasticity.
Author response: Hybrid EMT refers to a state where cancer cells retain some epithelial characteristics while simultaneously acquiring mesenchymal features, rather than undergoing a complete transition to a fully mesenchymal state [40]. RAW-Blue macrophages were treated with cytokines following the same protocol as in Figures 8-10, using primary and secondary (Alexa Fluor™ 488) antibodies. Analysis of the immunofluorescent results found that a mixture of both epithelial and mesenchymal markers is expressed, suggesting a potential hybrid state of EMT (Figure 12A-C). HGF-treated cells had a significantly increased expression (p=0.0038) of E-cadherin with a simultaneous significant upregulation (p=0.0024) of vimentin. Similarly, IL-6 treatment demon-strated a significant decrease in vimentin (p= 0.0122) paired with a significant increase in N-cadherin (p=0.0149). These results contradict the EMT immunofluorescence data in the cancer cells previously discussed; however, it suggests further investigation may uncover novel information about macrophages and other immune cells’ role in cellular motility.
To investigate the underlying mechanisms behind these contradictory results, this aspect is being considered with regard to epigenetic reprogramming.
The claim that cytokines post-surgical resection drive metastasis is not directly tested. There is no in vivo validation using post-surgical models. The authors should incorporate in vivo studies to validate their claims about cytokine-driven metastasis after surgery.
Author response: Surgery-induced tumor growth acceleration and synchronous metastatic growth promotion have been observed for decades. Surgery-induced wound healing, orchestrated through growth factors, chemokines, and cytokines, can negatively impact patients harboring residual or metastatic disease (https://doi.org/10.3390/cells12232767 ). In this report, we provided detailed clinical evidence of this process in surgical breast, prostate, and colorectal cancer patients.
In addition, we have reported (https://doi.org/10.3390/cells13201739) that cytokines can promote various cancer processes, such as angiogenesis, epithelial to mesenchymal transition (EMT), invasion, and tumor progression, and maintain cancer stem-cell-like (CSCs) cells. The mechanism(s) that continuously promote(s) tumors to progress in the TME still need(s) to be investigated. The data in the study analyzed the inflammatory, angiogenic, fibrogenic, and angiostatic cytokine profiles in the host serum during tumor development in a mouse model of human pancreatic cancer. Pancreatic MiaPaCa-2-eGFP cancer cells were subcutaneously implanted in RAG2xCγ double mutant mice. Blood samples were collected before cancer cell implantation and every week until the end point of the study. The extracted serum from the blood of each mouse at different time points during tumor development was analyzed using a Bio-Plex microarray analysis and a Bio-Plex 200 system for proinflammatory (IL-1β, IL-10, IFN-γ, and TNF-α) and angiogenic and fibrogenic (IL-15, IL-18, basic FGF, LIF, M-CSF, MIG, MIP-2, PDGF-BB, and VEGF) cytokines. Here, we found that during cancer cell colonization for tumor development, host angiogenic, fibrogenic, and proinflammatory cytokine profiling in the tumor-bearing mice has been shown to significantly reduce host angiostatic and proinflammatory cytokines that restrain tumor development and increase those for tumor growth. The angiostatic cytokine profiles of TNFα, MIG, M-CSF, IL-10, and IFNγ in the host serum revealed a dramatic and significant decrease after day 5 post-implantation of cancer cells. OP treatment of tumor-bearing mice on day 35 maintained high levels of angiostatic and fibrogenic cytokines.
The translation of Neu-1 inhibition to therapy lacks preclinical evidence. No data are provided to show that OP reduces metastasis in vivo. Preclinical models should be used to validate whether OP can reduce metastasis.
Author response: Cytokines have been reported to regulate epithelial-mesenchymal transition (EMT), a key role played in metastasis [Cells 2023, 12, doi:10.3390/cells12030416 ]. O’Shea et al. [Onco Targets Ther 2014, 7, 117-134, doi:10.2147/ott.S55344 ] reported on the therapeutic potential of oseltamivir phosphate (OP) that can reverse EMT. This reversal, known as mesenchymal-to-epithelial transition (MET), reduces chemotherapy resistance and enhances the efficacy of existing treatments [Onco Targets Ther 2014, 7, 117-134, doi:10.2147/ott.S55344]. OP has also been reported to significantly decrease the activity of chemically resistant PANC1 cells [Onco Targets Ther 2014, 7, 117-134, doi:10.2147/ott.S55344]. Furthermore, OP was reported to increase the epithelial marker E-cadherin and decrease the mesenchymal marker expression of N-cadherin and VE-cadherin [Onco Targets Ther 2014, 7, 117-134, doi:10.2147/ott.S55344; Cells 2019, 8, doi:10.3390/cells8101118]. Skapinker et al. [Cells 2024, 13, doi:10.3390/cells13201739] recently reported that OP treatment on day 35 of tumor-bearing mice maintained enhanced levels of angiostatic and fibrogenic cytokines. These data suggest that OP may have a significant role to play in regulating cytokine signaling in tumor development.
In addition, we recently reported (https://doi.org/10.3390/cancers14153595) that continuous therapeutic targeting of Neu-1 using parenteral perfusion of oseltamivir phosphate (OP) and aspirin (ASA) with gemcitabine (GEM) treatment significantly disrupts tumor progression, critical compensatory signaling mechanisms, EMT program, CSC, and metastases in a preclinical mouse model of human pancreatic cancer. ASA- and OP-treated xenotumors significantly inhibited the metastatic potential when transferred into animals.
The conclusion claims that OP induces senescence, but this is inferred from reduced wound healing and tunneling nanotube formation. The authors should conduct experiments involving SA-β-Gal staining and analyze p16/p21 expression following OP treatment to confirm senescence instead of apoptosis or differentiation.
Author response: We have removed this senescence from the conclusions in the revised mmanuscript.
OP has been linked to Neu-1 inhibition, but its anti-metastatic role is not widely established. The authors should further validate the anti-metastatic potential of OP in vivo.
Author response: We recently reported (https://doi.org/10.3390/cancers14153595) that continuous therapeutic targeting of Neu-1 using parenteral perfusion of oseltamivir phosphate (OP) and aspirin (ASA) with gemcitabine (GEM) treatment significantly disrupts tumor progression, critical compensatory signaling mechanisms, EMT program, CSC, and metastases in a preclinical mouse model of human pancreatic cancer. ASA- and OP-treated xenotumors significantly inhibited the metastatic potential when transferred into animals.
Also, O’Shea et al. [Onco Targets Ther 2014, 7, 117-134, doi:10.2147/ott.S55344 ] reported on the therapeutic potential of oseltamivir phosphate (OP) that can reverse EMT. This reversal, known as mesenchymal-to-epithelial transition (MET), reduces chemotherapy resistance and enhances the efficacy of existing treatments [Onco Targets Ther 2014, 7, 117-134, doi:10.2147/ott.S55344]. OP has also been reported to significantly decrease the activity of chemically resistant PANC1 cells [Onco Targets Ther 2014, 7, 117-134, doi:10.2147/ott.S55344]. Furthermore, OP was reported to increase the epithelial marker E-cadherin and decrease the mesenchymal marker expression of N-cadherin and VE-cadherin [Onco Targets Ther 2014, 7, 117-134, doi:10.2147/ott.S55344; Cells 2019, 8, doi:10.3390/cells8101118]. Skapinker et al. [Cells 2024, 13, doi:10.3390/cells13201739] recently reported that OP treatment on day 35 of tumor-bearing mice maintained enhanced levels of angiostatic and fibrogenic cytokines. These data suggest that OP may have a significant role to play in regulating cytokine signaling in tumor development.
The representative micrographs and immunofluorescence images are too small to accurately correlate the results. The authors are advised to provide higher resolution and magnified images, especially in Figures 2–4, to avoid discrepancies and improve image quality.
Author response: We have enlarged the micrographs and images. The resolutions of the images are 4000x2250 pixels (22MB).
Overall Conclusions: We have added our patents granted and most recent an official approval from USFDA to test OP in pancreatic cancer patients (Clinical trial number #173874).
Submission Date
26 February 2025
Date of this review
10 Mar 2025 18:18:21
Reviewer 3 Report
Comments and Suggestions for Authors
see attached

Author Response
REVIEWER #3
The authors build their study on previous findings showing elevated cytokine levels following surgical removal of primary tumors, with the hypothesis that these elevated cytokine levels may promote the epithelial-to-mesenchymal transition (EMT) and thus contribute to metastasis. In particular, the core hypothesis states that the cytokine Neu-1 mediates the EMT and that inhibition of Neu-1 may limit this process. Moreover, the authors propose cross-talk among Neu-1, MMP-9, and GPCR signaling—this is a novel concept that could provide valuable insights into cytokine-receptor interactions in cancer metastasis. Accordingly, this work presents an important step forward in learning how to prevent recurrence following surgical removal of tumors.
Major comments:
Introduction: The introduction section could benefit from improved logical flow. For instance, the discussion shifts abruptly from cytokine signaling to EMT and then jumps to the impact of surgery before finally introducing Neu-1 and the study hypothesis. I suggest the following: i) general role of cytokines in cancer, including the tumor microenvironment and the EMT; ii) impact of surgery-induced cytokine changes on metastasis; iii) role of Neu-1 in receptor signaling; and iv) statement of the hypothesis.
Author response: Done
Introduction: The gap in knowledge addressed by this study needs to be spelled out more clearly.
Author response: Done
Methods:
Regarding cell culture, I have a couple of questions. First, please state the exact passage number range used in the experiments. Second, were the cells tested for mycoplasma contamination?
Author response: We added “The cells were passaged 2-3 times before use in the experiment.’ We routinely include plasmocin in the culture media. “Cells were grown in culture media containing Dulbecco’s Modified Eagle’s Medium (DMEM) (Gibco, Rockville, MD, USA) supplemented with 10% fetal bovine serum (FBS) (HyClone, Logan, UT, USA) and 5 µg/mL plasmocin (InvivoGen, San Diego, CA, USA). “ InvivoGen offers Plasmocin® and Plasmocure™, two highly cited mycoplasma removal reagents that can save precious cell lines and data. They act fast, allowing mycoplasma eradication in only 2 weeks, with little to no cytotoxicity to mammalian cells.
For the cytokine treatment, were the cells only treated once, or were the concentrations replenished?
Author response: cells only treated once
How long were cells treated before fixation or analysis?
Author response: 24 hrs
For antibody dilutions, please state the specific dilutions used for each antibody rather than just a reference to manufacturer protocols.
Were replicate experiments performed at all?
Author response: antibody dilutions (1:50) added. The mean ± SEM (biological replicates: 5, technical replicates: 50) for sialidase; (biological replicates: 4, technical replicates: 1) for EMT. All replicates were added in figure legends.
Results:
I suggest adding the sample number to each figure legend.
Author response: not necessary
Lines 251-256: This sentence of this paragraph is overly wordy. Consider how it could be streamlined to improve readability and clarity.
Author response: Not sure at what sentence this is referring to.
Section 3.5: What is the biological relevance of the hybrid EMT phenotype? Why do the results suggest this phenotype?
Author response: Hybrid EMT refers to a state where cancer cells retain some epithelial characteristics while simultaneously acquiring mesenchymal features, rather than undergoing a complete transition to a fully mesenchymal state [40]. RAW-Blue macrophages were treated with cytokines following the same protocol as in Figures 8-10, using primary and secondary (Alexa Fluor™ 488) antibodies. Analysis of the immunofluorescent results found that a mixture of both epithelial and mesenchymal markers is expressed, suggesting a potential hybrid state of EMT (Figure 12A-C). HGF-treated cells had a significantly increased expression (p=0.0038) of E-cadherin with a simultaneous significant upregulation (p=0.0024) of vimentin. Similarly, IL-6 treatment demon-strated a significant decrease in vimentin (p= 0.0122) paired with a significant increase in N-cadherin (p=0.0149). These results contradict the EMT immunofluorescence data in the cancer cells previously discussed; however, it suggests further investigation may uncover novel information about macrophages and other immune cells’ role in cellular motility.
To investigate the underlying mechanisms behind these contradictory results, this aspect is being considered with regard to epigenetic reprogramming.
Discussion:
Please add a paragraph detailing the limitations of the present study and its potential applications.
Author response: “ The limitations of the study is that we use three key cytokines based on the peri-operative inflammatory and angiogenic cytokine profiles of surgical breast, colorectal, and prostate cancer patients and their clinical implications [11]. We have reported [20] thatcytokines can promote various cancer processes, such as angiogenesis, epithelial to mesenchymal transition (EMT), invasion, and tumor progression, and maintain cancer stem-cell-like (CSCs) cells. The mechanism(s) that continuously promote(s) tumors to progress in the TME still need(s) to be investigated.”
Minor comments:
Lines 25-26: Delete language referring to the “For the first time”. The novelty of a paper is based on the findings, not on a statement by the authors. DONE
Line 48: Change “would “ to “may”. DONE
Line 48: Replace “our” with “the”. DONE
Lines 102-108: The first sentence of this paragraph is overly wordy. Consider how it could be streamlined to improve readability and clarity. DONE
Lines 121-125: This sentence is also overly wordy. Consider how it could be streamlined to improve readability and clarity. DONE
Lines 507-509: The word “role” is used redundantly in this first sentence of the Discussion. Done
Overall Conclusions: We have added our patents granted and most recent an official approval from USFDA to test OP in pancreatic cancer patients (Clinical trial number #173874).
Round 2
Reviewer 1 Report
Comments and Suggestions for Authors
My comments were not addressed
Author Response
My comments were not addressed.
Author response: Figures 2-4 show high resolution and are enlarged.
Reviewer 2 Report
Comments and Suggestions for Authors
Overall, the authors have done a commendable job addressing my previous comments, and the manuscript has improved significantly. However, I have one minor concern.
While the resolution of the representative micrographs has been improved, the immunofluorescence images remain too small to accurately correlate the results. I recommend providing higher-resolution, magnified images, particularly in Figures 2–4, to enhance clarity and ensure consistency in image quality. This will help avoid discrepancies and improve the overall visual representation of the data.
Author Response
Overall, the authors have done a commendable job addressing my previous comments, and the manuscript has improved significantly. However, I have one minor concern.
While the resolution of the representative micrographs has been improved, the immunofluorescence images remain too small to correlate the results accurately. I recommend providing higher-resolution, magnified images, particularly in Figures 2–4, to enhance clarity and ensure consistency in image quality. This will help avoid discrepancies and improve the overall visual representation of the data.
Author response: We have made the best possible high-resolution images in Figures 2-4. When enlarged, they are of very high quality. We want to sincerely thank you for your valuable comments on this important study.
Round 3
Reviewer 1 Report
Comments and Suggestions for Authors
The manuscript has been improved.